# An ISR-independent role of GCN2 prevents excessive ribosome biogenesis and mRNA translation

Mónica Román-Trufero[1,2,3], Istvan T Kleijn[4], Kevin Blighe[5], Jinglin Zhou[2], Paula Saavedra-García[2], Abigail Gaffar[2], Marilena Christoforou[2], Axel Bellotti[1], Joel Abrahams[6], Abdelmadjid Atrih[7], Douglas Lamont[7], Marek Gierlinski[8], Pooja Jayaprakash[9], Audrey M Michel[9], Eric O Aboagye[6], Mariia Yuneva[3], Glenn R Masson[10], Vahid Shahrezaei[4], Holger W Auner[1,2,3,11]

**The integrated stress response (ISR) is a corrective physiological programme to restore cellular homeostasis that is based on the attenuation of global protein synthesis and a resource-enhancing transcriptional programme. GCN2 is the oldest of four kinases that are activated by diverse cellular stresses to trigger the ISR and acts as the primary responder to amino acid shortage and ribosome collisions. Here, using a broad multi-omics approach, we uncover an ISR-independent role of GCN2. GCN2 inhibition or depletion in the absence of discernible stress causes excessive protein synthesis and ribosome biogenesis, perturbs the cellular translatome, and results in a dynamic and broad loss of metabolic homeostasis. Cancer cells that rely on GCN2 to keep protein synthesis in check under conditions of full nutrient availability depend on GCN2 for survival and unrestricted tumour growth. Our observations describe an ISR-independent role of GCN2 in regulating the cellular proteome and translatome and suggest new avenues for cancer therapies based on unleashing excessive mRNA translation.**

## Introduction

Normal cellular function and tissue health rely upon the precise regulation of the cellular proteome to ensure the provision of specific proteins at ideal concentrations at all times (Vabulas & Hartl, 2005; López-Otín et al, 2013; Kaushik & Cuervo, 2015; Hipp et al, 2019; Kurtishi et al, 2019; Anisimova et al, 2020; Takemon et al, 2021). To maintain proteostasis, cells must coordinate mRNA translation and protein degradation, which in turn are balanced with metabolic processes that orchestrate the provision and use of macromolecular building blocks and energy. Under anabolic conditions, such as those driven by oncogenic signalling in cancer cells, the increase in global protein synthesis, which is tied to an increase in the biogenesis of ribosomal proteins, ribosomal RNA, and translation factors, places an additional burden on the cellular mechanisms of proteostasis (Kamphorst et al, 2015; Sullivan et al, 2019; Faubert et al, 2020; Pavlova et al, 2022; Finley, 2023).

The integrated stress response (ISR) is a fundamental homeostatic reaction and one of the pivotal mechanisms of cellular proteome regulation (Harding et al, 2003). In animals, four kinases couple different stress signals to the phosphorylation of the translation initiation factor eIF2α, which results in global attenuation of mRNA translation. At the same time, the preferential translation of the transcription factor ATF4 triggers a programme of resource optimisation that includes suppression of mTORC1, activation of autophagy, and increased amino acid synthesis and uptake (Tallóczy et al, 2002; Harding et al, 2003; B'chir et al, 2013; Ye et al, 2015; Bröer et al, 2016). GCN2 is the oldest of the four eIF2α kinases and is conserved in all eukaryotes. GCN2-mediated ISR signalling modulates a range of biological processes such as circadian physiology (Pathak et al, 2019), learning and memory (Costa-Mattioli et al, 2005), haematopoietic and skeletal stem cell function (Hu et al, 2020; Li et al, 2022), hepatic red blood cell clearance and iron metabolism (Toboz et al, 2022), and antitumour immunity (Halaby et al, 2019). Although GCN2 is activated in response to diverse stresses, the best-studied role of GCN2 is in the cellular adaptation to amino acid deficiency, where GCN2 is activated by uncharged tRNAs (Wek et al, 1989, 1995; Dong et al, 2000). GCN2 activation also occurs upon ribosome stalling as a consequence of translation elongation problems and requires the interaction of GCN2 with the ribosomal P stalk (Wek et al, 1995; Dong et al, 2000; Harding et al, 2000; Castilho et al, 2014; Ishimura et al, 2016; Inglis et al, 2019; Masson, 2019; Wu et al, 2020). In line with their high

[1]Division of Haematology and Central Haematology Laboratory, Lausanne University Hospital (CHUV), Lausanne, Switzerland [2]Hugh and Josseline Langmuir Centre for Myeloma Research, Department of Immunology and Inflammation, Imperial College London, London, UK [3]The Francis Crick Institute, London, UK [4]Department of Mathematics, Imperial College London, London, UK [5]Clinical Bioinformatics Research, London, UK [6]Department of Surgery and Cancer, Imperial College London, London, UK [7]FingerPrints Proteomics Facility, School of Life Sciences, University of Dundee, Dundee, UK [8]Data Analysis Group, Division of Computational Biology, School of Life Sciences, University of Dundee, Dundee, UK [9]EIRNA Bio Ltd, Cork, Ireland [10]Division of Cancer Research, School of Medicine, University of Dundee, Dundee, UK [11]Faculty of Biology and Medicine, University of Lausanne, Lausanne, Switzerland

Correspondence: holger.auner@chuv.ch

demand for amino acids to sustain proliferation and growth, GCN2 promotes cancer cell survival under conditions of nutrient scarcity (Ye et al, 2010; Parzych et al, 2019; Cordova et al, 2022; Missiaen et al, 2022; Nofal et al, 2022). GCN2 also protects haematopoietic cancer cells from deleterious effects of asparaginase treatment and proteasome inhibition, two commonly used anti-cancer therapies that trigger intracellular amino acid depletion (Suraweera et al, 2012; Nakamura et al, 2018; Saavedra-García et al, 2021). We have recently shown that a subset of tumours of diverse tissue origins share transcriptional signatures with cancer cell lines whose survival is dependent on GCN2 without suffering evident stress (Saavedra-García et al, 2021). Together with recent observations that GCN2 supports the proliferation of colon cancer cells under nutrient-rich conditions (Piecyk et al, 2024), the observations indicate that GCN2 regulates important cellular functions in an ISR-independent manner.

Using an integrated systems-level approach that includes transcriptome, proteome, translatome, and metabolome profiling, we show that in a subset of cancer cells, GCN2 prevents excessive ribosome biogenesis and protein synthesis under conditions of optimal nutrient availability. This function of GCN2 is distinctly different from and independent of its canonical role in the ISR, is largely regulated on a translational level, and is required for unimpeded tumour growth and cancer cell survival and for maintaining metabolic homeostasis.

# Results

## GCN2 maintains the cancer cell transcriptome and proteome independently of the ISR

According to the Cancer Dependency Map (DepMap) portal, ~8% of cancer cell lines are dependent on GCN2 based on CRISPR screening (https://www.proteinatlas.org/humanproteome/tissue/druggable). To confirm the deleterious effect of GCN2 inactivation in dependent cell lines, we treated 16 different solid tumour cell lines of varying degrees of GCN2 dependency, as well as four different types of primary healthy cells: CD34+ haematopoietic stem cells, mesenchymal stromal cells, human umbilical vein endothelial cells (HUVECs), and human dermal fibroblasts (HDFs), with the well-characterised GCN2 inhibitor, GCN2iB (Nakamura et al, 2018; Saavedra-García et al, 2021; Missiaen et al, 2022). We confirmed that GCN2 inhibition triggers cell death in cell lines defined as GCN2-dependent by DepMap under conditions of optimal nutrient availability, but not in those with low GCN2 dependency, or in the healthy primary cells we tested (Fig 1A). Based on their response to 48 h of 1 $\mu$M GCN2iB treatment, cancer cell lines were classified into three categories: highly dependent cells, which showed greater than 50% cell death; dependent cells, which exhibited 10–50% cell death; and independent cells with less than 10% death after treatment. We further confirmed GCN2 dependency by knocking down GCN2 in 3 cell lines (Figs 1B and S1A–C). Thus, survival of a subset of cancer cells depends on GCN2 in the absence of nutrient depletion or other stressors known to activate GCN2 or the ISR. To study the mechanistic basis for this potentially ISR-independent

role of GCN2, we first carried out transcriptome analysis by RNA sequencing (RNA-seq) in the melanoma cell line A375 as a model for GCN2-dependent cells. Using gene set enrichment analysis (GSEA) (Korotkevich et al, 2021 Preprint), we identified "MYC targets" and "E2F targets" as the most positively enriched categories (Fig 1C), indicating that GCN2 inhibition results in the activation of a transcriptional programme conducive to cell growth and proliferation despite its detrimental effect on viability. On the contrary, we observed that major metabolic pathways such as "glycolysis" and "fatty acid metabolism" were negatively enriched upon GCN2 inhibition (Fig 1C). Many of the genes included in the Hallmark term "MYC targets" are involved in proteostasis control, including genes coding for ribosomal and proteasomal subunits and translation initiation factors. To further explore how GCN2 regulates the proteome under conditions of nutrient abundance, we therefore carried out quantitative proteome profiling of GCN2iB-treated A375 cells. Our GSEA of differentially expressed proteins was largely concordant with the results of the transcriptome analysis, confirming "MYC targets" and "E2F targets" as the most enriched and "glycolysis" and "fatty acid metabolism" as suppressed pathways (Fig 1D). Notable pathways with discordant involvement between transcriptomic and proteomic responses included "Unfolded Protein Response" and "protein secretion," which were induced at the mRNA level but repressed at the protein level (Fig 1C and D). In line with these results, Gene Ontology (GO) analyses of both the transcriptome and proteome data showed a dominant up-regulation of mRNAs and proteins involved in ribosome and ribonucleoprotein complex biogenesis, whereas processes linked to extracellular matrix structure and organisation were repressed (Fig S1D).

We next inhibited or knocked down GCN2 in xenografted A375 cells to study its role in vivo. In the pharmacological model, mice were injected with A375 cells and treated with GCN2iB for 10 d from the time when tumours became palpable. In the genetic model, we xenografted A375 cells carrying a doxycycline-inducible shRNA targeting GCN2 (Fig S1E). With these approaches, we observed a reduction in tumour growth of 24% and 23%, respectively (Fig S1F and G). Although only the reduction in tumour growth in response to inhibitor treatment was statistically significant, the effect of GCN2 knockdown was numerically comparable. Taken together, the results provide strong support for the notion that GCN2 is required for unimpeded tumour growth in vivo. Moreover, GSEA of RNA-seq data from tumour sample mRNA showed a transcriptome response to prolonged in vivo GCN2 inhibition or depletion that was highly concordant with our findings in cultured cells after short-term pharmacological inhibition, including the induction of "MYC targets" and the repression of "glycolysis" (Fig 1E and F).

Intrigued by the fact that we did not observe enrichment in ISR-related pathways in response to GCN2 inhibition or depletion, we then directly compared the effect of GCN2 inhibition on A375 cells that were growing under conditions of optimal nutrient availability with the response in cells that were depleted of glutamine. Although GCN2iB widely suppressed the induction of ATF4 targets that was triggered by glutamine depletion, it had no such effect on the expression of key ISR transcripts in cells that were not deprived of glutamine (Figs 2A and S2A). Moreover, GCN2iB blocked the increase

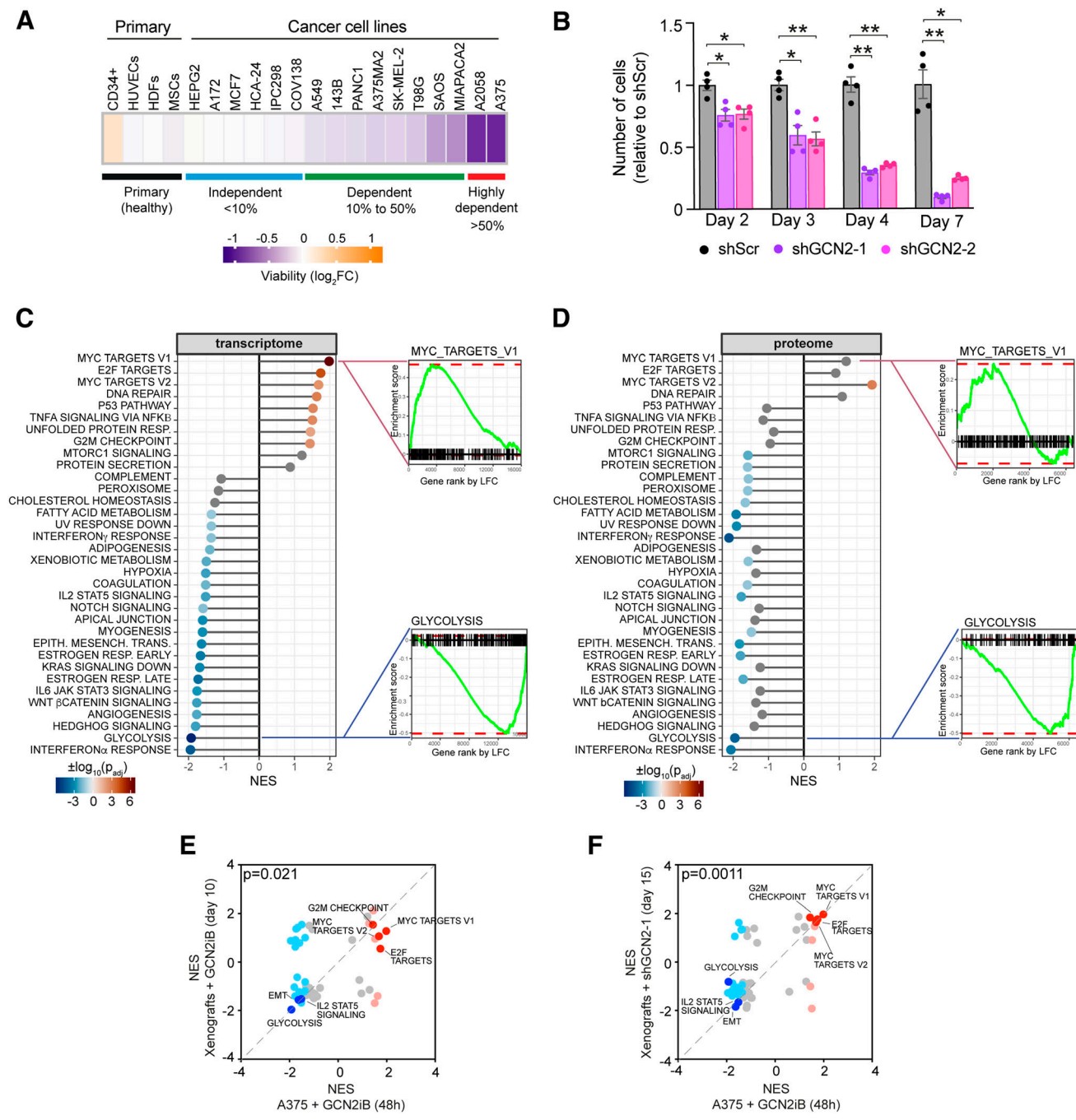

**Figure 1. GCN2 regulates the cancer cell proteome and transcriptome.**
**(A)** Heatmap showing the mean change in viability in the indicated primary cells and cell lines. Data are expressed as $\log_2$ GCN2iB-treated/DMSO control cells (cell lines, HDFs, and HUVECs, n = 3; CD34[+], n = 2; mesenchymal stromal cells, n = 4). The cancer cell line dependency classification is based on the indicated percentage of cell death after 48 h of GCN2iB treatment at 1 $\mu$M. **(B)** Viability of A375 cells after GCN2 knockdown. Data are shown as the mean ± SEM of four independent experiments. Statistical significance was determined by two-way ANOVA and Sidák's test for multiple comparisons. *$P < 0.05$, **$P < 0.005$. **(C)** GSEA of mRNA-level changes in A375 cells treated with GCN2iB (1 $\mu$M, 48 h), as determined by RNA-seq, shown as lollipop plots indicating normalised enrichment scores, and enrichment plots of selected Hallmark categories. **(D)** GSEA of protein abundance changes in A375 cells treated with GCN2iB, as determined by TMT labelling proteomics, shown as lollipop plots indicating normalised enrichment scores (NES), and enrichment plots of selected Hallmark categories. Lollipop colours in (C, D) indicate the degree of statistical significance (grey = not significant). **(C, E)** Correlation of NES from GSEA of RNA-seq data from cultured cells treated with GCN2iB as described in (C) and from tumour samples after 10 d of oral GCN2iB treatment. **(C, F)** Correlation of NES from GSEA of RNA-seq data from cultured cells treated with GCN2iB as described in (C) and tumour samples after 15 d of doxycycline-induced shRNA expression. **(E, F)** Red and blue dots indicate significantly (adjusted $P < 0.05$) up-regulated and down-regulated Hallmark categories. Dark red and blue highlight selected categories of interest.

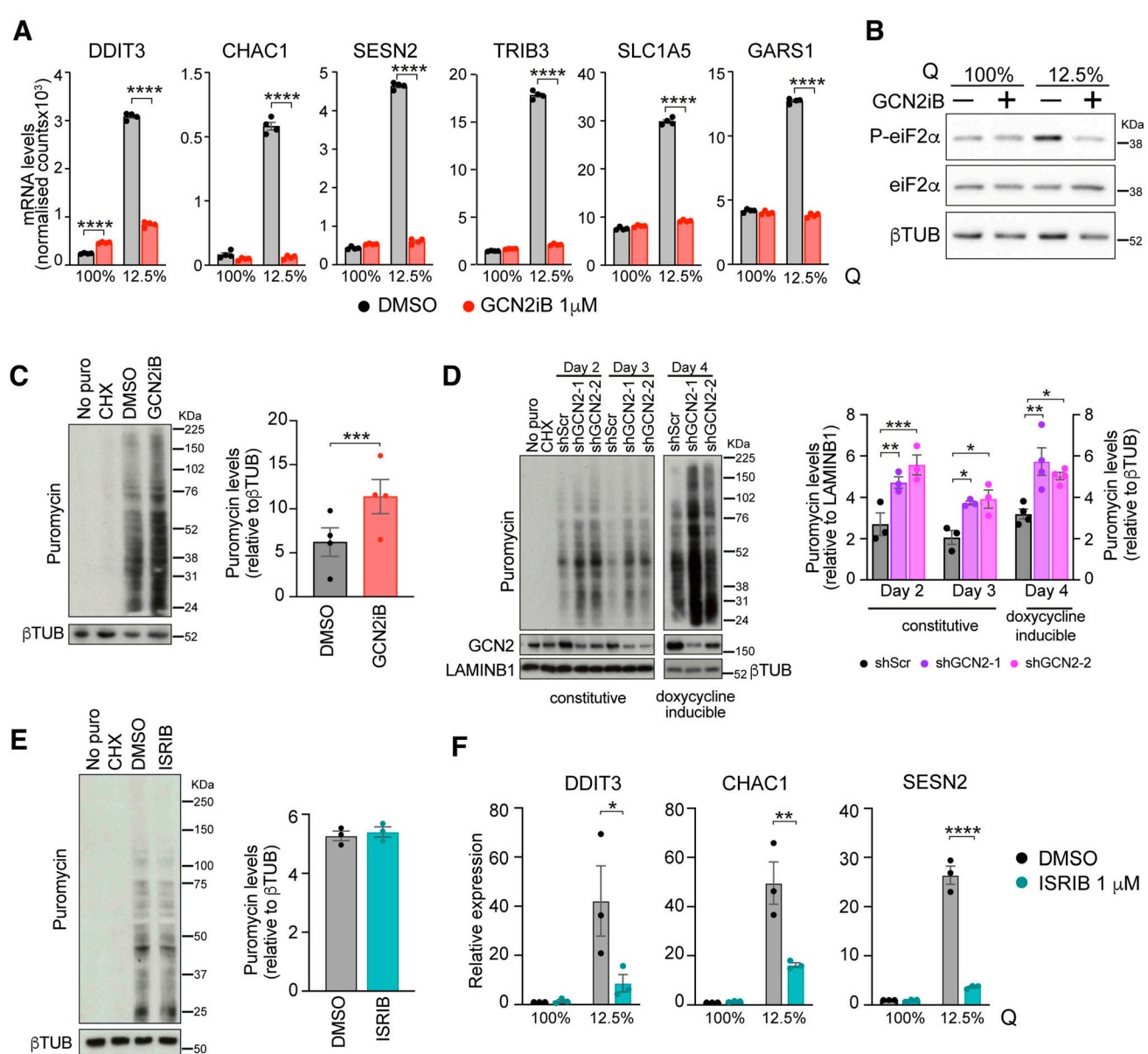

**Figure 2. GCN2 restrains translation independently of the ISR.**
**(A)** mRNA expression levels as determined by RNA-seq of selected ATF4 targets in A375 cells cultured in complete medium (2 mM glutamine, 100%Q) or glutamine-depleted medium (250 nM glutamine, 12.5%Q) and treated with GCN2iB (1 μM, 48 h). Data are shown as the mean ± SEM of four independent experiments. Statistical significance was determined by two-way ANOVA and Sidák's test for multiple comparisons. **(A, B)** Immunoblot analysis of the indicated proteins in A375 cells cultured and treated as described in (A). **(C)** Immunoblot analysis of puromycinylated proteins in A375 cells treated for 48 h with 1 μM GCN2iB (left), and bar graphs showing the quantification of puromycinylated proteins (right). Data are shown as the mean ± SEM of four independent experiments. Statistical significance was determined by a paired *t* test. **(D)** Left panel, immunoblot analysis of puromycinylated proteins in A375 cells expressing the indicated constitutive (n = 3) or doxycycline-inducible shRNAs (n = 4). Right panel, bar charts showing the quantification of puromycinylated proteins. Data are shown as the mean ± SEM. Statistical significance was determined by two-way ANOVA and Tukey's test for multiple comparisons. **(C, D)** No puro, extract from A375 cells grown without puromycin; CHX, treatment with 200 μg/ml cycloheximide for 3 h before puromycin addition. **(E)** Immunoblot analysis of puromycinylated proteins in A375 cells treated for 48 h with 1 μM ISRIB (left), and bar graphs showing the quantification of puromycinylated proteins (right). Data are shown as the mean ± SEM of three independent experiments. **(F)** mRNA levels of selected ISR genes in A375 cells grown in complete and glutamine-depleted medium and in the presence of 1 μM ISRIB for 48 h. mRNA levels were determined by qRT–PCR; data are shown as the mean ± SEM, n = 3. Statistical significance was determined by two-way ANOVA and Sidák's test for multiple comparisons. *$P < 0.05$, **$P < 0.005$, ***$P < 0.001$, ****$P < 0.0001$. Source data are available for this figure.

in eIF2α phosphorylation that was triggered by glutamine depletion but had no effect on eIF2α phosphorylation under conditions of full glutamine availability (Fig 2B). Thus, the function of GCN2 in cells that do not suffer from nutrient depletion is distinctly different

from its role as a central ISR regulator in response to amino acid scarcity.

Our observations so far indicate that GCN2 inhibition or depletion leads to an up-regulation of the protein biosynthesis

machinery both at the mRNA and at the protein level. To determine whether GCN2 is indeed required to keep protein synthesis in check, we performed a puromycin incorporation assay, which showed a 1.8-fold increase in protein synthesis in GCN2iB-treated cells (Fig 2C). We then quantified protein synthesis in A375 cells expressing shRNAs against GCN2 under a constitutive or a doxycycline-inducible promoter. The results show that a 45–70% reduction in GCN2 protein levels led to a 1.6–2.1-fold increase in global protein synthesis (Fig 2D).

To further verify that the cell death and the observed increase in protein synthesis after GCN2 inhibition are not linked to ISR inhibition, we cultured A375 cells in the presence of ISRIB, a well-characterised ISR inhibitor (Sidrauski et al, 2013). ISRIB had no discernible impact on cell viability at concentrations up to 5 $\mu$M (Fig S2B). Similarly, treatment with 1 $\mu$M ISRIB in complete medium did not result in increased translation (Fig 2E). Furthermore, ISRIB did not alter the expression of key ISR components in cells grown in complete medium although it effectively suppressed their up-regulation under amino acid deprivation (Fig 2F). Lastly, inhibition of the ISR had no significant effect on ribosome biogenesis at the mRNA level (Fig S2C).

Given the increased protein synthesis resulting from GCN2 inactivation, we then assessed the activity of mTORC1, a key regulator of protein synthesis that can be inhibited by GCN2 under nutrient-poor conditions (Ye et al, 2015). As indicated by lower levels of phosphorylation of S6K, we observed the expected mTORC1 repression after glutamine depletion, which was blocked by GCN2 inhibition. Under conditions of optimal nutrient availability, we observed no difference in S6K phosphorylation as a surrogate marker for mTORC1 activity after 24 or 48 h of GCN2 inhibition (Fig S2D), despite the increase in protein synthesis at that time point. Thus, GCN2 curtails global protein synthesis in some cancer cells growing under conditions of unrestricted nutrient availability without changes in ISR signalling or mTORC1 activity.

We then conducted transcriptome analysis on five additional cancer cell lines with different degrees of GCN2 dependency (see Fig 1A), as well as HUVECs to represent primary healthy cells, in which GCN2 was inhibited. We observed that GCN2-independent cell lines clustered differently from GCN2-dependent lines and primary cells (Fig S3A) and that the most dependent cell lines (A375, MiaPaCa2, and A375MA2) clustered together, whereas the least dependent cell line (A549) clustered close to independent cell lines (HepG2 and IPC298) and primary cells (HUVECs). Moreover, clustering based on groups of translation-related genes revealed up-regulation of transcripts coding for ribosomal proteins, MYC targets, and translation initiation factors in GCN2-dependent but not in GCN2-independent cells or primary cells (Fig S3B–E). Furthermore, analysis of ribosomal genes in mRNA levels in HUVECs and HDFs after GCN2iB treatment revealed that GCN2 inhibition does not increase ribosome biogenesis at the mRNA level in primary cells (Fig S4A and B).

We also carried out puromycinylation assays in another highly dependent cell line (A2058), one dependent cell line (MiaPaCa2), two GCN2-independent cell lines (IPC298 and HepG2), and two types of primary cells (HUVECs and HDFs) and observed evidence for increased protein synthesis in A2058 and MiaPaCa2 cells under conditions of GCN2 inhibition, whereas no changes were observed in IPC298, HepG2 cells, or the primary cells (Fig S5A). Moreover, as we

had observed in A375 cells, we did not observe changes in the expression of ATF4 targets indicative of ISR activation in any of the cells after GCN2 inhibition under conditions of full nutrient availability (Fig S5B). Finally, we carried out a proteomic analysis of two GCN2-independent cancer cell lines and HUVECs after GCN2 inhibition and observed that GCN2 inhibition had essentially no effect on the proteome of these cells (Fig S6A–D).

## GCN2 inactivation has differential effects on the transcriptional and proteomic regulation of fundamental cellular processes

Next, we analysed the extent to which different cellular processes are regulated by GCN2 at the transcriptome or proteome level. We found a positive and significant but overall moderate correlation between the changes in mRNA and protein abundance in GCN2iB-treated cells (Fig 3A). Although 71% of differentially expressed genes and proteins (DEGPs) changed concordantly (both mRNA and protein levels increased or decreased) in response to GCN2iB, 29% showed a discordant response (Fig 3B). To better understand this differential regulation, we divided DEGPs into five groups depending on whether mRNA and protein levels were unchanged, increased, or decreased, and categorised proteins according to their cellular localisation (Uhlén et al, 2015) (Fig 3B). This analysis revealed important differences between the 5 DEGP groups. Arguably, the biggest difference we observed was that the proportion of secreted proteins was twice as high in the concordantly repressed compared with the concordantly induced category (Fig 3B), a finding that is consistent with the repression of extracellular matrix–related pathways described above (Fig 1C). We then carried out a KEGG pathway enrichment analysis of the 5 DEGP categories (Fig 3B). Although metabolic and extracellular matrix–related pathways dominated the concordantly repressed category (Fig 3B, bottom left panel), RNA processing and transport, and ribosomal pathways topped the concordantly induced category (Fig 3B, upper right panel). "Ribosome" was also the top enriched pathway in the category of DEGPs marked by decreased mRNA but increased protein levels (Fig 3B, upper left panel). The DEGP category of reduced protein and increased mRNA levels (Fig 3B, bottom right panel) was dominated by "proteasome," which governs the coordinated breakdown of most cellular proteins. Plotting DEGPs belonging to significantly enriched GO terms on the mRNA/protein correlation diagram further confirmed and illustrated the differential transcriptomic and proteomic effects of GCN2 inhibition on ribosome biogenesis, proteasome subunits, and the extracellular matrix (Fig S7A–C). These results suggest a central role of GCN2 in the differential regulation of the transcriptome and proteome of cancer cells. Specifically, inhibition of GCN2 up-regulates processes linked to protein synthesis, especially ribosome biogenesis, whereas key metabolic pathways and extracellular matrix–related processes are predominantly repressed. Considering that most drug targets are proteins, we then used our proteomic data to investigate the effects of GCN2 inhibition on the druggable proteome of A375 cells. The results show that GCN2 inhibition significantly deregulated 484 out of 1,053 druggable proteins listed in the Human Protein Atlas (https://www.proteinatlas.org/humanproteome/tissue/druggable

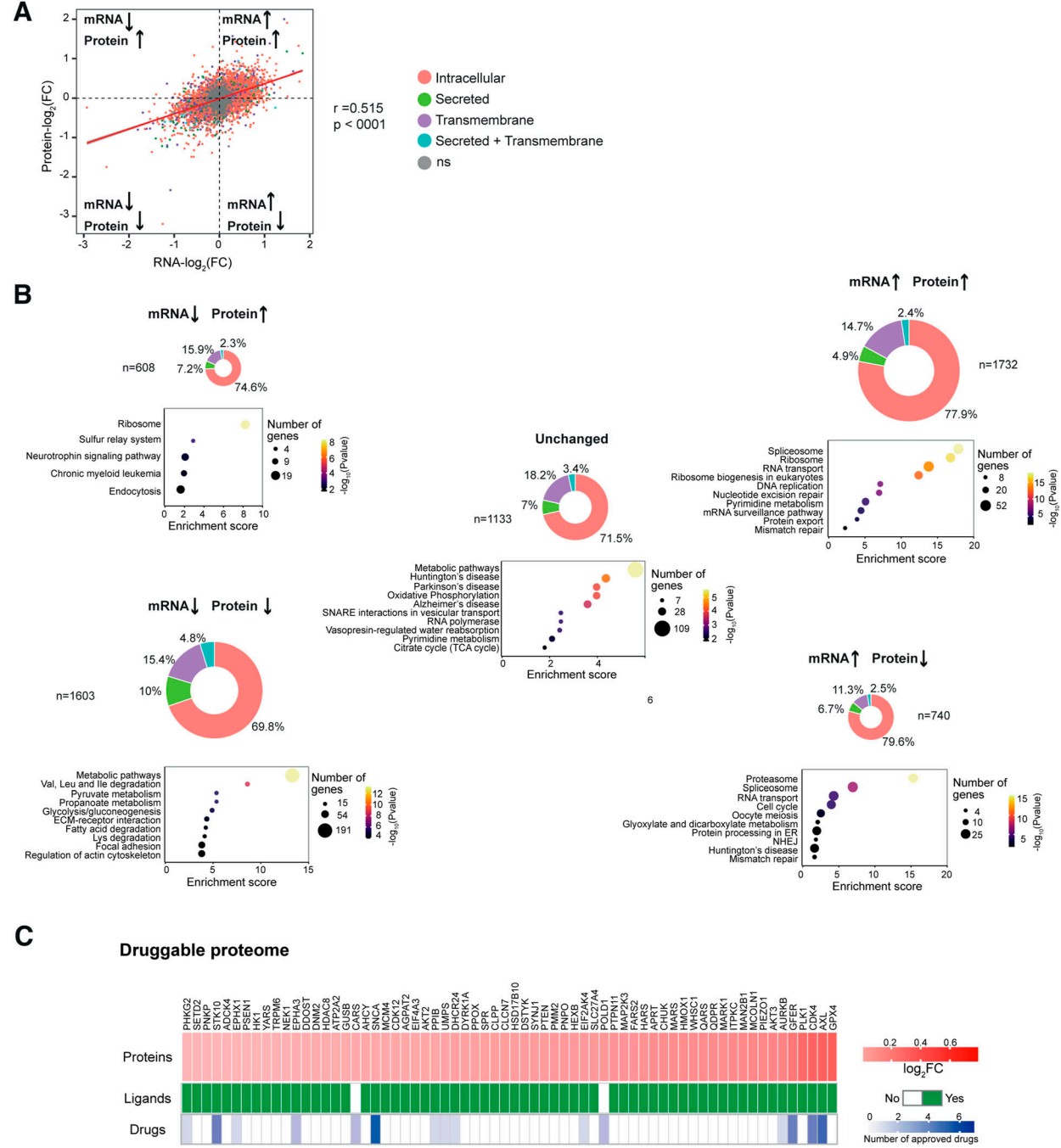

**Figure 3. Differential regulation of the transcriptome and proteome by GCN2.**
**(A)** Correlation analysis of mRNA and protein expression changes in A375 cells treated with 1 μM GCN2iB for 48 h. Each dot represents a transcript and corresponding protein. Colours indicate cellular localisation of proteins, and grey dots represent transcripts and proteins with non-significant (ns) changes in mRNA and protein abundance. **(B)** Doughnut charts depict the proportion of intracellular, secreted, and transmembrane proteins. Bubble plots show the top 10 enriched KEGG pathways with bubble sizes indicating the number of genes and bubble colours representing statistical significance levels. **(C)** Heatmap representation of GCN2iB-induced protein expression changes of 66 up-regulated proteins (top) for which at least one active preclinical ligand (middle) or approved drug (bottom) is registered in ChEMBL.

accessed on 28th July 2023). Of these, 306 were down-regulated and 178 were up-regulated. The latter include 66 proteins for which active ligands are known and 15 for which one or more approved drugs exist (Fig 3C). Thus, GCN2 inhibition deregulates the cellular proteome, thereby potentially altering its druggability.

## GCN2 restrains ribosome biogenesis

To understand the mechanisms underlying the differential regulation of the proteome by GCN2, we performed ribosome profiling (Ribo-seq), a deep sequencing approach based on the

proposition that the density of ribosome-protected fragments (RPFs) of a particular transcript provides an indication of the rate of protein synthesis (Ingolia et al, 2009). First, using standard quality control analyses, we confirmed that RPFs showed the anticipated size distribution and predominantly mapped to annotated coding regions (Fig S8A and B). Given that GCN2 has been shown to play a key role in the resolution of ribosome collisions (Ishimura et al, 2016; Wu et al, 2020), we then investigated whether GCN2 inhibition caused an increase in ribosome stalling. However, our data showed no significant changes in codon occupancy (Fig 4A) or in the pause score (Kumari et al, 2018) (Fig S8C), indicating that GCN2 is not required for the prevention or resolution of ribosome collisions in A375 cells that are not depleted of nutrients. Next, we investigated whether inhibition of GCN2 alters translation efficiency (TE), which is a measure of the number of ribosomes that occupy a specific transcript and is defined as the ratio of RPFs to mRNA abundance. We identified 604 genes with significantly different TE (differential TE genes, DTEGs) in GCN2iB-treated cells. Of these, 55.4% exhibited increased TE, indicating a significant increase in ribosome occupancy and thus translation, whereas 44.6% showed reduced TE (Fig 4B). Moreover, the median shift in TE for genes exhibiting increased TE was significantly greater than that observed for genes with decreased TE (Fig S8D), in line with our observation that GCN2 inactivation increases global protein synthesis (Fig 2C and D). Furthermore, GO enrichment analysis of DTEGs revealed that translation- and ribosome-related terms dominated the most enriched pathways (Fig S8E).

We then further subcategorised DTEGs in line with a previously published classification (Chothani et al, 2019), according to their transcriptional or translational regulation. The "exclusive" group consists of transcripts whose change in TE is driven by increased or decreased ribosome occupancy with no change in mRNA levels (n = 249 genes), whereas the "intensified" group (n = 91) is composed of transcripts whose change in abundance is accompanied by a concordant increase or decrease in ribosome occupancy. Transcripts with "buffered" regulation (n = 264) show changes in ribosome occupancy that offset the change in mRNA abundance. "Forwarded" transcripts are characterised by mRNA changes that are not accompanied by a change in TE (n = 2,307) (Fig 4C). Subsequent GO analysis of the genes included in these categories showed that processes linked to the regulation of translation and ribosome biogenesis dominated the exclusive and intensified categories and showed some enrichment in the buffered and forwarded categories (Fig 4D). Remarkably, 91.2% (93 out of 102) of ribosomal protein mRNAs exhibited higher ribosome occupancy in GCN2iB-treated cells (Figs 4E and S8F), indicating that they were translated more actively. Thus, GCN2 regulates the expression of ribosomal proteins on the transcriptional and even more so on the translational level.

Given that protein synthesis relies on proteasome function to degrade proteins and recycle amino acids (Vabulas & Hartl, 2005; Suraweera et al, 2012), we analysed Ribo-seq data of proteasome subunits and detected no change in the TE of genes encoding proteasomal subunits (Fig S9A). This finding adds to our transcriptome and proteome analyses, which showed an increase in proteasome subunit mRNAs but the lower level of proteasomal proteins (Figs 3B and S7). Thus, the increase in global protein synthesis that is unleashed by GCN2 inhibition and linked to increased ribosome biogenesis is not accompanied by an increase in proteasome biogenesis. Given that we had observed repression of ECM-related pathways on both the transcript and the protein level (Fig 3B), we also inspected our Ribo-seq data for changes in ECM mRNA translation and found reduced ribosome occupancy for most of the fibrous ECM proteins, proteoglycans, and integrins (Fig S9B). Thus, GCN2 keeps global mRNA translation in check but is required to maintain ECM protein synthesis.

To visualise the relation between processes that are regulated by GCN2 on a translational level, we carried out a network analysis of enriched Reactome pathways. The results of this analysis highlight that pathways directly involved in proteome control, such as rRNA processing, translation, post-translational protein modifications, protein localisation, and protein degradation, are closely interconnected with other key cellular functions, notably metabolic and cell cycle–related processes (Fig S9C).

## GCN2 inhibition triggers a dynamic loss of metabolic homeostasis

Given the substantial energy and resource demands of protein synthesis, we then set out to determine the metabolic sequelae of GCN2 inhibition. We first quantified intracellular ATP levels in GCN2iB-treated live cells and found them to be significantly reduced by 11% (Fig 5A). We then carried out time-resolved untargeted metabolome. Our data revealed a widespread disturbance of cell metabolism with 113 (13.1%), 209 (24.2%), and 298 (34.5%) of 865 detected metabolites significantly deregulated after 12, 24, and 36 h of GCN2 inhibition, respectively (Figs 5B and S10A). Amino acids and lipids were the most deregulated metabolic families at each time point. Specifically, we found 16 out of 20 proteogenic amino acids to be significantly reduced (Fig 5C), an observation compatible with greater demand during heightened translation. An in-depth analysis of the lipid superpathway showed evidence of augmented fatty acid catabolism together with a substantial increase in sphingosines, phospholipids, and ceramides (Fig 5D), lipids that are integral components of cell membranes. In conjunction with the transcriptional and translational up-regulation of components of the ER translocon and signal peptidase complexes (Fig S10B), these findings may be indicative of a cellular attempt to expand the ER to accommodate enhanced protein folding.

Finally, we analysed processes involved in energy generation. Despite finding no changes in glucose levels, we observed a reduction in the concentration of downstream metabolites in the glycolytic pathway that was most pronounced at the 36-h time point (Fig 5E), in line with our transcriptome, proteome, and translatome data (Figs 1C and D and S10C). On the contrary, TCA cycle components remained largely unchanged (Fig S10D).

Taken together, the data demonstrate that GCN2 inhibition causes a dynamic and broad loss of metabolic homeostasis that culminates in ATP deficiency.

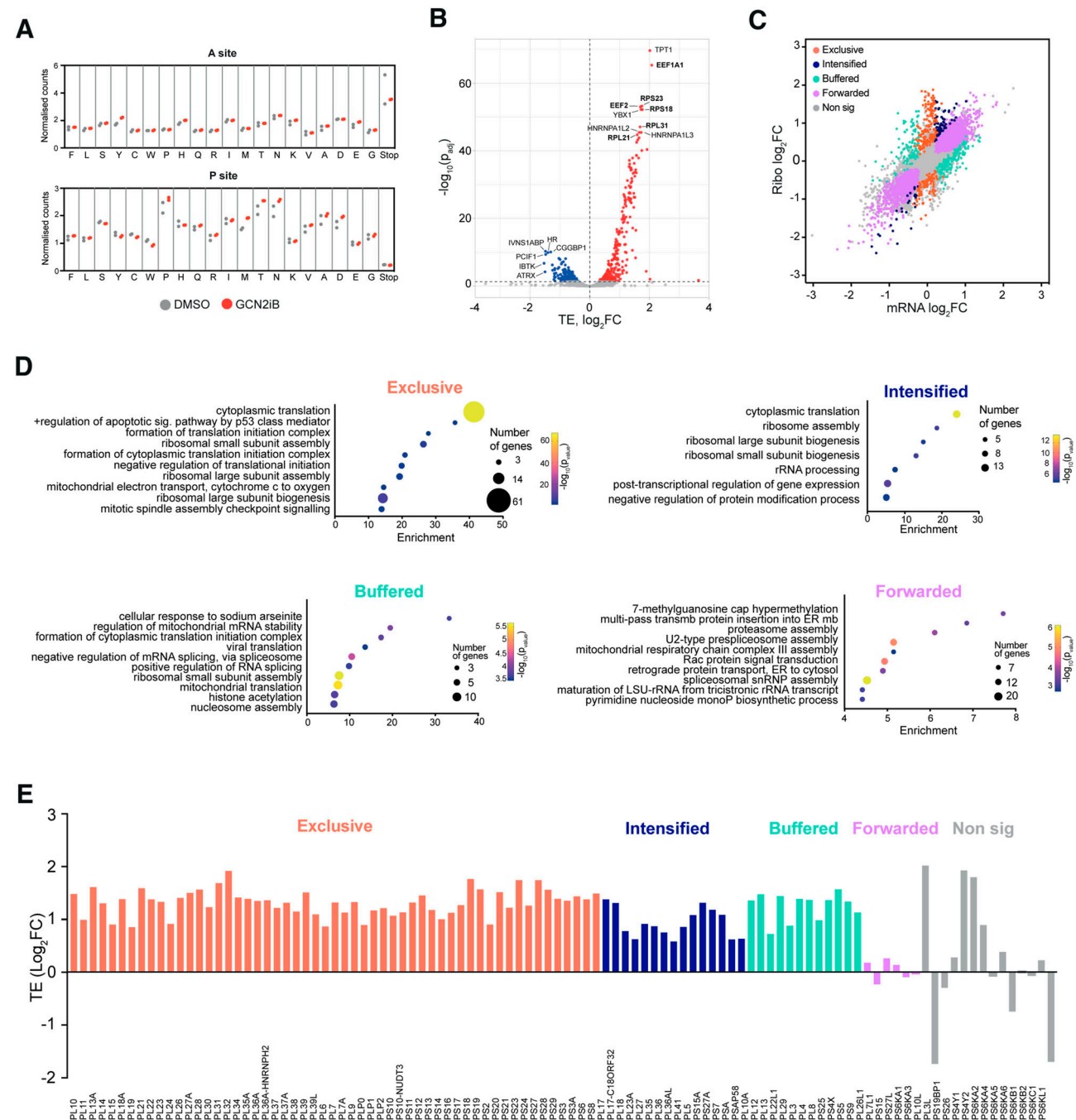

**Figure 4. GCN2 controls translation of ribosomes and translation factors.**
**(A)** Codon occupancy at ribosomal A and P sites as determined by Ribo-seq in A375 cells treated with 1 μM GCN2iB for 48 h (n = 2). Dots show normalised RPF counts for the indicated amino acids and stop codons. **(B)** Volcano plot showing genes with differential translation efficiency (TE) in A375 cells treated with GCN2iB (1 μM, 48 h). Labels indicate genes with the lowest adjusted *P*-value in the up-regulated and down-regulated groups. Labels in the bold font indicate genes directly involved in protein translation. **(C)** Scatter plot of changes between GCN2iB-treated cells compared with DMSO-treated controls in Ribo-seq data (y-axis) and paired RNA-seq data (x-axis). Each dot represents a gene and is coloured according to its regulatory grouping as indicated. **(C, D)** Bubble plots show the most enriched Biological Process (BP) Gene Ontology (GO) terms in the groups depicted in (C). **(E)** Bar chart showing changes in TE of ribosomal subunit transcripts in GCN2iB-treated cells.

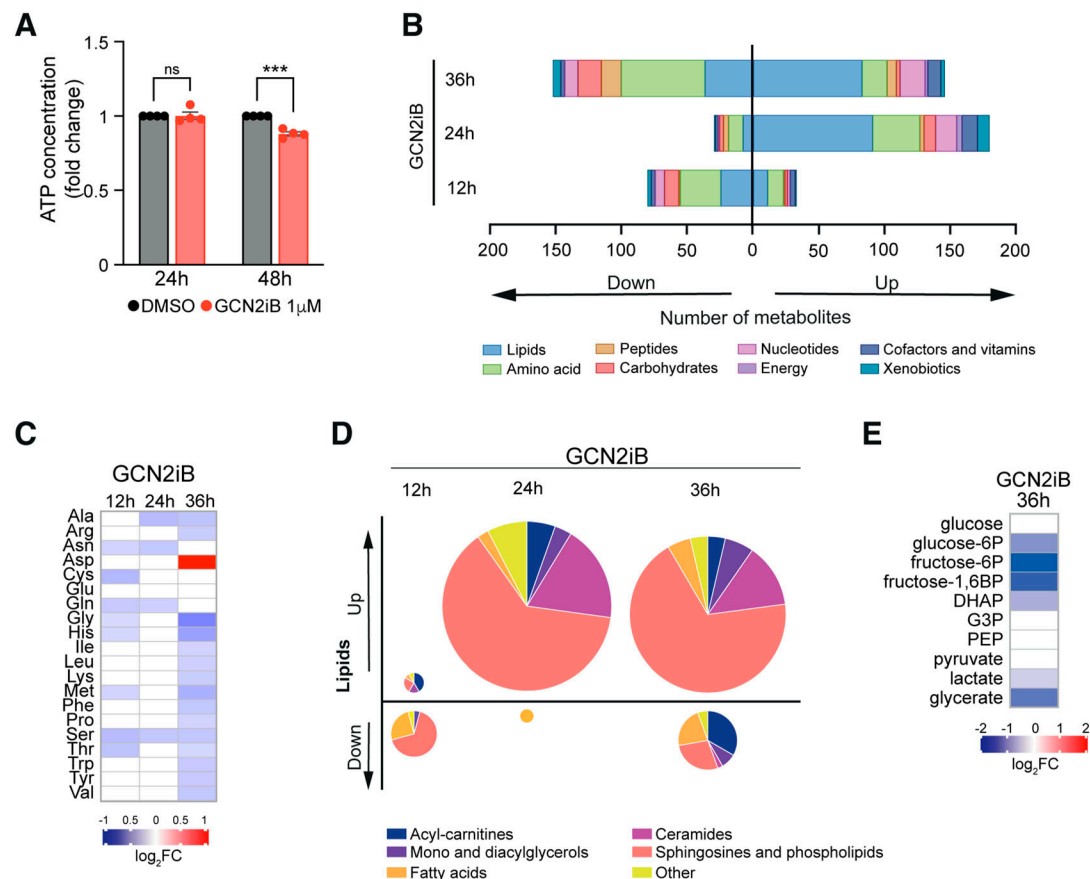

**Figure 5. Loss of metabolic homeostasis upon GCN2 inhibition.**
**(A)** Bar chart showing the quantification of intracellular ATP levels as determined by a luciferase-based assay and normalised to the number of live cells. Data are shown as the mean ± SEM of four independent experiments. Statistical significance was determined by two-way ANOVA and Sidák's test for multiple comparisons, ***$P <$ 0.001. **(B)** Number and classification of deregulated metabolites, quantified by ultra-high-performance liquid chromatography–tandem mass spectrometry (UHPLC-MS/MS). **(C)** Heatmap showing the relative concentration of proteogenic amino acids. **(D)** Distribution of deregulated lipid families, with a pie chart size representing the absolute number of deregulated lipids. **(E)** Heatmap showing the relative concentration of glycolysis intermediates. **(C, E)** Data are shown as the mean intensity normalised to DMSO-treated cells of four independent experiments. All panels show results in A375 cells cultured in complete medium and treated with 1 µM GCN2iB for the indicated times.

# Discussion

The cellular functions of GCN2 in healthy and cancerous cells have largely been elucidated under stress conditions such as amino acid deficiency, where GCN2 triggers the ISR, an adaptive signalling network that responds to proteostasis perturbations. Here, using cancer cell lines with different degrees of GCN2 dependency as experimental models, we show that GCN2 plays a key role in regulating the cellular proteome in a manner that is independent of the ISR (Figs 2A, B, and E and S2A and B), results that are supported by our observations that the ISR inhibitor ISRIB had no relevant effect on the viability or transcriptome of GCN2-dependent cells (Figs 2F and S2B and C). These observations, together with our findings on cells in which GCN2 was genetically depleted, also rule out the possibility that the GCN2 inhibitor we used in our study activated the ISR (Carlson et al, 2023; Szaruga et al, 2023). Thus, the role of GCN2 as a regulator of translation in non-stressed cells, which we describe here, differs significantly from its role as a translation regulator in the ISR. Although stress-induced GCN2

activation in the canonical ISR attenuates steady-state translation via eIF2α phosphorylation, our analyses show that at least in certain cells, GCN2 prevents excessive translation in the absence of nutrient depletion or other known stress conditions. Translatome analyses show that this is accompanied and likely enabled by the up-regulation of ribosome biogenesis. However, it remains unknown exactly how the depletion or inhibition of GCN2 triggers this response. In the absence of changes in ISR signalling, it can be assumed that what we observe does not simply constitute the silencing of an ISR that has been activated at basal growth conditions by an unknown stress. It is worth noting that a role of GCN2 in nutrient-rich conditions has recently been described for colorectal cancers. However, in contrast to our work, GCN2 inhibition was observed to trigger a reduction in protein synthesis and mTORC1 activity (Piecyk et al, 2024). Thus, the role of GCN2 in regulating translation and the cellular proteome appears to be highly cell- and context-dependent. Moreover, transcriptomic and proteomic analyses of primary cells treated with GCN2iB suggest that this role of GCN2 is not functionally relevant in non-transformed cells (Figs S3, S4, S5, and S6). It remains to be determined why the ISR-independent role

of GCN2 in preventing excessive translation is restricted to certain cancer cells.

In cancers, increased protein synthesis as part of an enhanced anabolic programme to sustain tumour growth is tied to an increase in the biogenesis of ribosomal proteins, ribosomal RNA, and translation factors to enhance translation capacity (Iritani & Eisenman, 1999; Inoki et al, 2002; Ma et al, 2005; Truitt & Ruggero, 2016) and is typically under the control of hyperactive oncogenes such as MYC. Indeed, anabolic programmes driven by MYC have been shown to trigger an adaptive stress response controlled by GCN2 and ATF4 (Tameire et al, 2019; Croucher et al, 2021). These observations are compatible with the notion that a MYC-driven increase in protein synthesis requires a higher level of coordination between amino acid use and availability, thereby making cells more reliant on the ISR and GCN2. However, this does not provide an explanation for the induction of MYC targets by GCN2 inactivation, unless GCN2 has a hitherto unknown inhibitory effect on MYC in some cancer cells. To our knowledge, no such link is known. Moreover, the observation that GCN2 inactivation is detrimental to certain cancer cells while at the same time leading to the induction of MYC targets and protein anabolism, typically linked to tumour growth, is intriguing. However, MYC can trigger apoptosis in cells where growth signals are insufficient (Askew et al, 1991; Evan et al, 1992, 1994).

For some time, the only known target of GCN2 kinase activity was eIF2α. However, recent research has provided increasing evidence for the existence of additional targets (Dokládal et al, 2021; Ge et al, 2023; Stonyte et al, 2023). It is therefore possible that the ISR-independent roles of GCN2 that we observe are mediated by one of the known or other yet to be defined GCN2 targets. In addition, despite being principally a cytosolic kinase, GCN2 can localise to the nucleolus (Nakamura & Kimura, 2017) suggesting not only a possible explanation for a direct role in ribosome biogenesis, but also the possibility of hitherto unknown interactions with nuclear proteins. It is therefore tempting to speculate that at least in some cancers, GCN2 has a direct or indirect inhibitory effect on MYC, potentially through regulation of one of MYC's multiple partners (Lourenco et al, 2021). Such an interaction could have important implications for GCN2 in the physiological regulation of MYC (Eilers & Eisenman, 2008) and for anti-cancer therapies, particularly those targeting strongly MYC-driven tumours.

One of the questions arising from our observations is how excessive protein synthesis triggered by GCN2 inactivation could be detrimental to at least some cancer cells. Mechanistically, our observations provide several potential explanations for how this occurs. First and foremost, it is important to note that mRNA translation is a highly resource-intensive and energy-consuming process that uses between a quarter and almost half of cellular ATP supplies (Buttgereit & Brand, 1995; Princiotta et al, 2003). Thus, the approximate doubling of global protein synthesis we observed upon GCN2 inhibition or knockdown can be expected to substantially enhance metabolic demands. Indeed, the dynamic and broad loss of metabolic homeostasis we observed is compatible with the notion that the excessive protein production that was triggered by GCN2 inactivation exceeded the adaptive capacity of cells. Our findings are also compatible with a recently proposed counterintuitive approach to cancer therapy that is based on the deliberate overactivation of oncogenic signals to overwhelm the stress response capacity of cancer cells (Dias et al, 2024).

One other likely important issue is the failure of cells to enhance proteasome capacity in parallel with the increase in ribosome biogenesis and protein synthesis. This is an important problem as the use of amino acids to produce new proteins must be balanced with increased provision through protein degradation, a key role of the proteasome. Indeed, protein synthesis upon nutrient restriction relies on the proteasome (Vabulas & Hartl, 2005), and proteasome inhibition rapidly causes intracellular amino acid scarcity even when extracellular amino acid supplies are plentiful (Suraweera et al, 2012; Parzych et al, 2019; Saavedra-García et al, 2021). It is therefore reasonable to assume that the combination of excessive translation and insufficient proteasomal protein degradation accounts for the decline in intracellular amino acid levels that we observed. Because amino acids not only serve as macromolecular building blocks for proteins but are also converted into energy (DeBerardinis et al, 2008), it is likely that this decrease in intracellular amino acids also contributed to the decrease in ATP levels we observed. The reduction in glycolysis we observed may also hinder the maintenance of energy homeostasis in cells.

Approximately one third of cellular proteins are membrane-bound or destined for secretion. These proteins must be post-translationally modified and folded in the ER (Dobson, 2004) and include ECM proteins, such as collagens, which are particularly resource-intensive to produce. It was therefore intriguing to observe that ECM proteins featured prominently among those suppressed by GCN2 inactivation on a transcript, protein, and translational level. Given that tumours that produce ATP at a slow rate suppress protein synthesis to cut down on energetically expensive functions (Bartman et al, 2023), it is reasonable to hypothesise that the production of proteins that are particularly resource-intensive was reduced because of lacking resources. It is worth noting that we observed an induction of ER-related pathways in cells in which GCN2 was inhibited, particularly in some translocation channels (Figs 1C and S10B). Together with our metabolomic data that show a substantial increase in lipids that are typically found in membranes, the results suggest a cellular response directed at expanding the secretory apparatus. However, we did not find convincing evidence of ER stress, a condition characterised by the excessive accumulation of misfolded proteins in the ER (Walter & Ron, 2011), suggesting that the secretory pathway was not overwhelmed by the enhanced protein synthesis that was unleashed by GCN2 inactivation.

Our observations reveal a hitherto unrecognised function of GCN2 in curtailing protein synthesis and ribosome biogenesis that is independent from its canonical role in the ISR. They add to the growing body of evidence that the proteome and its regulation at the translational level play central but understudied roles in cancer biology and lend support to the emerging concept of therapeutic overactivation of oncogenic signalling as a potential cancer treatment strategy.

# Materials and Methods

### Cell culture

The source and culture conditions for the cell lines used in this work are detailed in Table S1.

CD34[+] haematopoietic stem cells and mesenchymal stromal cells, as well as the research ethics approval for their use, have been previously described (Loaiza et al, 2018). Briefly, CD34[+] cells and mesenchymal stromal cells were sourced from the Imperial College Healthcare Tissue and Biobank (ICHTB, Human Tissue Authority licence 12275). ICHTB is approved by the UK National Research Ethics Service to release human material for research (12/WA/0196).

## GCN2 knockdown in A375 cells

Plasmids and sequences used for constitutive shRNA-mediated GCN2 knockdown have been described previously (Parzych et al, 2019). Oligonucleotides with the same sequences were cloned into an EZ-Tet-pLKO vector (Frank et al, 2017) (85966; Addgene). Lentiviral particles were produced by transfecting HEK293 cells with a shRNA-carrying plasmid and the envelope and packaging vectors (pMD2.G, pRSV.REV, pMDLg/pRRE, a kind gift from Prof. Anastasios Karadimitris, Imperial College London) using Lipofectamine 2000 (11668030; Invitrogen), and virus-containing supernatants were collected 48 and 72 h after transfection. A375 cells were transduced by spinoculation with 500 $\mu$l of the lentivirus-containing supernatant. For induction of shRNA expression with doxycycline, 500 ng/ml of doxycycline (D9891; Sigma-Aldrich) was added to the cell culture medium.

## GCN2 inhibition

Cells were incubated for the indicated times with 1 $\mu$M of GCN2 inhibitor GCN2iB (HY-112654; MedChemExpress) or vehicle control (DMSO, D2650; Sigma-Aldrich), and medium was changed every 24 h.

## Cell viability assay

Cell viability was determined using CellTiter 96 AQueous Non-Radioactive Cell Proliferation Assay (MTS) (G5430; Promega), following the manufacturer's instructions. Absorbance was measured in a FLUOstar Omega (BMG Labtech) or a Tecan Spark multiplate reader.

## ATP measurement

ATP quantification was carried out with ATP Bioluminescence Assay Kit CLS II (11699695001; Roche) following the manufacturer's instructions. Luminescence was measured using a Tecan Infinite M200 microplate reader.

## Puromycinylation assay

For semi-quantitative analysis of protein synthesis, puromycin (P8833; Sigma-Aldrich) was added to cell cultures at a final concentration of 5 $\mu$g/ml for 10 min, after which cells were collected for protein extraction and Western blotting as described below.

## Protein extraction and Western blot

Cell pellets were resuspended in ice-cold RIPA buffer (R0287; Sigma-Aldrich) supplemented with phosphatase and protease inhibitors (PhosSTOP, 4906845001, and cOmplete EDTA-free, COEDTAF-RO; Roche) and incubated on ice for 20 min. Lysates were then cleared by centrifugation at 14,000$g$ for 10 min at 4°C. The protein concentration was measured with the Pierce BCA protein assay kit (23225; Thermo Fisher Scientific) according to the manufacturer's instructions. Extracts were diluted with 6x SDS loading buffer, boiled for 4 min, and subjected to SDS–PAGE, and gels were transferred to PVDF membranes using the Bio-Rad Mini-PROTEAN Tetra electrophoresis and wet-blotting system. Membranes were blocked with 5% BSA in 0.05% NP-40/Tris-buffered saline (TBS/NP-40) for 1 h at RT and incubated with the primary antibody diluted in 0.5% BSA/TBS/NP-40 overnight at 4°C. The next day, membranes were washed three times for 15 min at RT with TBS/NP-40, incubated with the appropriate secondary antibody diluted in 0.5% BSA/TBS/NP-40, washed three more times for 15 min at RT, and developed using Immobilon Crescendo Western HRP substrate (WBLUR0500; Millipore). Images were obtained with a Hyperfilm-ECL film (GE28906839; Cytiva) or iBright CL750 (Invitrogen). The antibodies and working dilutions used in this study are as follows: anti-P-eIF2$\alpha$ (1:1,000, 9721; Cell Signaling), anti-eIF2$\alpha$ (1:2,000, 9722; Cell Signaling), anti-puromycin (1:5,000, MABE343; Merck Millipore), anti-GCN2 (1:2,000, 3302; Cell Signaling), anti-lamin B1 (1:2,000, 12586; Cell Signaling), anti-$\beta$-tubulin (1:3,000, 2146; Cell Signaling), anti-P-S6 kinase (1:2,000, 9S34; Cell Signaling), anti-S6 kinase (1:2,000, 2708; Cell Signaling), goat anti-rabbit-HRP (1:10,000, A16096; Thermo Fisher Scientific), and rabbit anti-mouse-HRP (1:10,000, A16160; Thermo Fisher Scientific). Western blot quantification was performed with Fiji software (Schindelin et al, 2012).

## Tumour xenografts

### Genetic depletion of GCN2

All animal experiments were performed in accordance with the United Kingdom Home Office Guidance on the Operation of the Animals (Scientific Procedures) Act 1986 Amendment Regulations 2012 and within the published National Cancer Research Institutes Guidelines for the welfare and use of animals in cancer research (Workman et al, 2010). Experiments were conducted under Project Licence Number PP1780337. A total of 1.5 × 10[6] A375 cells in 100 $\mu$l PBS were injected into the rear flank of female NOD SCID Gamma mice. When tumours reached quantifiable size, mice were administered 0.5 mg/ml doxycycline (D9891; Sigma-Aldrich) in 0.5% sucrose water for 15 d. Tumour measurements were performed with callipers every 2 d, and mouse weight was monitored throughout the experiment. Tumour volume (TV) was calculated as TV = length (mm) × width (mm) × depth (mm) × $\prod/6$.

### Pharmacological inhibition of GCN2

5 × 10[6] A375 cells in 200 $\mu$l PBS with Matrigel (1:1) were injected into female BALB/c nude mouse flanks. When tumours reached 125–150 mm[3], animals were randomised into treatment and control groups. Mice were then treated with 10 mg/kg GCN2iB (or 0.5% methylcellulose vehicle control) by oral gavage twice a day, for 10 d. Mice were weighed, and tumour dimensions were measured with callipers twice a week. TV was calculated as TV = 0.5 × (long diameter) × (short diameter)[2], in mm.

## RNA extraction

RNA extraction from cultured cells was performed with the ReliaPrep RNA Miniprep Systems kit (Z6012; Promega) according to the manufacturer's instructions. RNA from xenograft tumours was extracted using TRIzol (15596026; Invitrogen) and cleaned with the RNeasy mini kit (74104; QIAGEN) following the manufacturer's protocols.

## Quantitative real-time polymerase chain reaction (qRT–PCR)

1 $\mu$g RNA was used for cDNA synthesis using GoScript Reverse Transcription System (Promega) following the manufacturer's guidelines. 2 $\mu$l of a 1:20 dilution of the synthesised cDNA was used for qRT–PCR using GoTaq qPCR Master Mix (Promega) in a QuantStudio 6 qPCR system (Thermo Fisher Scientific). mRNA levels were quantified using the ΔCt method, using HPRT as a control for normalisation. The primers used in this work are listed in Table S2.

## RNA sequencing

RNA samples were quantified using Qubit 4.0 Fluorometer (Life Technologies), and RNA integrity was checked with the RNA kit on Agilent 5300 Fragment Analyzer (Agilent Technologies). RNA-sequencing libraries were prepared using NEBNext Ultra II RNA Library Prep Kit for Illumina (NEB) following the manufacturer's instructions. Briefly, mRNAs were first enriched with oligo(dT) beads. Enriched mRNAs were fragmented for 15 min at 94°C. First-strand and second-strand cDNAs were subsequently synthesised. cDNA fragments were end-repaired and adenylated at 3' ends, and universal adapters were ligated to cDNA fragments, followed by index addition and library enrichment by limited-cycle PCR. Sequencing libraries were validated using NSG Kit on Agilent 5300 Fragment Analyzer and quantified by using Qubit 4.0 Fluorometer. The sequencing libraries were multiplexed and loaded on the flow cell on the Illumina NovaSeq 6000 instrument according to the manufacturer's instructions. The samples were sequenced using a 2 × 150 Pair-End (PE) configuration v1.5. Image analysis and base calling were conducted by NovaSeq Control Software v1.7 on the NovaSeq instrument. Raw sequence data (.bcl files) generated from Illumina NovaSeq were converted into fastq files and demultiplexed using Illumina bcl2fastq program version 2.20. One mismatch was allowed for index sequence identification. RNA library preparation and sequencing were conducted by Azenta.

## Tandem mass tag (TMT) labelling proteomics

### S-Trap processing of samples

80 × 10$^6$ cells per sample were collected, washed twice with ice-cold PBS supplemented with phosphatase and protease inhibitors (PhosSTOP, 4906845001, and cOmplete EDTA-free, COEDTAF-RO; Roche), and snap-frozen in liquid nitrogen. Cell pellets were lysed in 3 ml lysis buffer (100 mM TEAB, 5% SDS) and sonicated for 15 s three times to shear DNA, and protein concentration was estimated using the micro-BCA assay. Aliquots of 500 $\mu$g (two batches for each sample) of protein were processed using S-Trap

mini protocol (ProtiFi) as recommended by the manufacturer with little modifications. Protein's disulphide bonds were first reduced in the presence of 20 mM DTT for 10 min at 95°C, then alkylated in 40 mM IAA for 30 min in the dark. After sample application into an S-Trap mini spin column, trapped proteins were washed 5 times (500 $\mu$l) with S-Trap binding buffer. Double digestion with trypsin (Pierce; Thermo Fisher Scientific) (1:40) was carried out first overnight at 37°C in 160 $\mu$l 50 mM TEAB, and then for another 6 h at the same temperature. Elution of peptides from S-Trap mini columns was achieved by centrifugation at 1,000$g$ for 1 min after the addition of 160 $\mu$l 50 mM TEAB, then 160 $\mu$l 0.2% aqueous formic acid, and finally 160 $\mu$l 50% (vol/vol) acetonitrile containing 0.2% (vol/vol) formic acid. The resulting tryptic peptides were pooled and dried a couple of times in SpeedVac by resuspension/drying in 50 $\mu$l 100 mM TEAB or Milli-Q until the pH was around 8.5 (checked by pH strips). Peptides were then quantified using Pierce Quantitative Fluorometric Peptide Assay (Thermo Fisher Scientific).

### TMT labelling and high-pH reversed-phase fractionation

Tryptic peptides (200 $\mu$g, each sample) were dissolved in 200 $\mu$l of 150 mM TEAB. TMT labelling was performed according to the manufacturer's instructions (Thermo Fisher Scientific). The different TMT-16 plex labels (1 mg) (Thermo Fisher Scientific) were dissolved in 40 $\mu$l of anhydrous acetonitrile, and each label was added to a different sample. The mixture was incubated for 1 h at RT, and the labelling reaction was stopped by adding 8 $\mu$l of 5% hydroxylamine per sample. After labelling with TMT, samples were checked for labelling efficiency, then mixed, desalted, and dried in SpeedVac at 30°C. Samples were redissolved in 200 $\mu$l ammonium formate (10 mM, pH 9.5), and peptides were fractionated using high-pH RP chromatography. A C18 column (XBridge Peptide BEH, 130 Å, 3.5 $\mu$m, 2.1 × 150 mm; Waters) with a guard column (XBridge, C18, 3.5 $\mu$m, 2.1 × 10 mm; Waters) was used on Ultimate 3000 HPLC (Thermo Fisher Scientific). Buffers A and B used for fractionation consist, respectively, of (A) 10 mM ammonium formate in Milli-Q water, pH 9.5, and (B) 10 mM ammonium formate, pH 9.5, in 90% acetonitrile. Fractions were collected using a WPS-3000FC autosampler (Thermo Fisher Scientific) at 1-min intervals. The column and guard column were equilibrated with 2% Buffer B for 20 min at a constant flow rate of 0.2 ml/min. Fractionation of TMT-labelled peptides was performed as follows: 190 $\mu$l aliquots were injected onto the column, and the separation gradient was started 1 min after the sample was loaded onto the column. Peptides were eluted from the column with a gradient of 2% Buffer B to 20% Buffer B in 6 min, then from 20% Buffer B to 45% Buffer B in 51 min, and finally from 45% Buffer B to 100% Buffer B within 1 min. The column was washed for 15 min in 100% Buffer B. The fraction collection started 1 min after injection and stopped after 80 min (total 80 fractions, 200 $\mu$l each). Formic acid (30 $\mu$l of 10% stock) was added to each fraction and concatenated in groups of 20 fractions for total proteome.

### LC-MS analysis

Mass spectrometry data were collected using an Orbitrap Eclipse mass spectrometer (Thermo Fisher Scientific) coupled to Dionex UltiMate 3000 RS (Thermo Fisher Scientific). LC buffers used are the following: Buffer A (0.1% formic acid in Milli-Q water [vol/vol]) and Buffer B (80% acetonitrile and 0.1% formic acid in Milli-

Q water [vol/vol]). For total proteome analysis, an equivalent of 1 µg of each fraction was loaded at 15 µl/min onto a trap column (100 µm × 2 cm, PepMap nanoViper C18 column, 5 µm, 100 Å; Thermo Fisher Scientific) equilibrated in 0.1% TFA. The trap column was washed for 6 min at the same flow rate with 0.1% TFA and then switched in-line with a Thermo Fisher Scientific resolving C18 column (75 µm × 50 cm, PepMap RSLC C18 column, 2 µm, 100 Å), which was equilibrated at 3% Buffer B for 19 min at a flow rate of 300 nl/min. The peptides were eluted from the column at a constant flow rate of 300 nl/min with a linear gradient from 10% buffer to 18% within 83 min, from 18% B to 27% Buffer B in 45 min, and then from 27% B to 90% Buffer B within 5 min. The column was then washed with 90% Buffer B for 8 min. The column was kept at a constant temperature of 50°C.

The Orbitrap Eclipse mass spectrometer (Thermo Fisher Scientific) was operated in a positive ionisation mode, equipped with an easy spray source. The source voltage was set to 2.5 Kv, and the ion transfer tube was set to 275°C.

The scan sequence began with an MS1 spectrum (Orbitrap analysis; resolution 120,000; mass range 380–1,500 m/z; RF lens was set to 30%, AGC target was set to standard, maximum injection time was set to Auto, microscan 1, Monoisotopic peak determination was set to peptide, intensity threshold $5 \times 10^3$, charge state 2–6, data dependant mode was set to cycle time, time between master scans was set to 3 s, and exclusion duration was set to 60 s). MS1 spectra were acquired in a profile mode.

MS2 analysis consisted of CID and was carried out as follows: isolation mode quadrupole, isolation window 0.7, collision energy mode fixed, CID collision energy 30%, CID activation time 10 ms, activation Q 0.25, normalised AGC target was set to standard, maximum injection time 50 ms, microscan was set to 1, detector type ion trap, and mass range 400–1,200.

After acquisition of each MS2 spectrum, we collected an MS3 spectrum using the following parameters: number of SPS precursor was set to 10, MS isolation window 0.7, MS2 isolation window was set to 2, activation type HCD, HCD collision energy 55%, Orbitrap resolution 50,000, scan range 100–500, normalised AGC target 400%, maximum injection time 120 ms, and microscan was set to 1. Data for both MS2 and MS3 were acquired in centroid mode. The real-time search feature was active during the analysis using uniprot-proteome _up000005640.fasta.

## Ribosome profiling

Stranded mRNA-seq and ribosome profiling (Ribo-seq) libraries were generated by EIRNA Bio (https://eirnabio.com) from flash-frozen cell pellets. Cell pellets were lysed in ice-cold polysome lysis buffer (20 mM Tris, pH 7.5, 150 mM NaCl, 5 mM MgCl₂, 1 mM DTT, 1% Triton X-100) supplemented with cycloheximide (100 µg/ml). For stranded mRNA-seq, total RNA was extracted from 10% of lysate using TRIzol, before mRNA was poly(A)-enriched, fractionated, and converted into Illumina-compatible cDNA libraries. For Ribo-seq, the remaining lysates were digested in the presence of 35U RNase 1 for 1 h. After RNA purification and size selection of ribosome-protected mRNA fragments on 15% urea–PAGE gels, contaminating rRNA was

depleted from samples using EIRNA Bio's custom biotinylated rRNA depletion oligos before the enriched fragments were converted into Illumina-compatible cDNA libraries. Both stranded mRNA-seq libraries and Ribo-seq libraries were sequenced using 150PE on Illumina's NovaSeq 6000 platform to depths of 20 million and 100 million raw read pairs per sample, respectively.

## Untargeted metabolomic profiling

A375 cells were treated with 1 µM GCN2iB (or DMSO as a vehicle control) for 12, 24, and 36 h, with a medium change at 24 h. At each time point, cells were collected by trypsinisation and washed twice in PBS, and cell pellets were snap-frozen in liquid nitrogen. Untargeted metabolic profiling was performed by Metabolon, Inc with the Global Discovery HD4 panel, which uses ultra-high-performance liquid chromatography–tandem mass spectrometry (UHPLC-MS/MS). Sample preparation, mass detection, quantification, and initial data analysis were carried our following the Metabolon, Inc protocols described previously (Evans et al, 2009, 2012, 2014; DeHaven et al, 2010). Further pathway analysis was performed with MetaboAnalyst 5.0 (www.metaboanalyst.ca).

## Bioinformatics analyses

### RNA sequencing

**Raw data processing** Read count abundance was generated from raw data FASTQ files by pseudo-aligning these to the *Homo sapiens* GENCODE "comprehensive" reference transcriptome (GRCh38.p12/release 39) using Salmon v1.6.0 (Patro et al, 2017). Transcript isoform-level counts were then imported to R Programming Language (R) (R Core Team, 2016) and summarised to gene-level counts (adjusting for gene length) using the tximport package in R.

Protein-coding genes were then isolated from the raw counts based on the GENCODE biotype keyword "protein_coding." Genes with zero counts across all samples were removed. The raw counts of the remaining genes were then converted to a DESeq2 (Love et al, 2014) object for normalisation. For downstream analyses, the negative binomial–distributed normalised counts were converted to regularised log (rlog) expression levels via the rlog function of DESeq2 in R. Variance-stabilised expression levels were also generated.

**Differential expression analysis** Differential expression analysis was conducted on the negative binomial–distributed normalised counts with an FDR set at 5%. After differential expression, log (base 2) fold changes (log₂FC) were shrunk via the lfcshrink function of DESeq2. A gene was defined as differentially expressed if it passed the Benjamini–Hochberg q ≤ 0.05.

**ATF4/DDIT3 targets** ChIP-seq targets of ATF4 and CHOP were extracted from a mouse study (Han et al, 2013) and converted to human orthologues using the Mouse Genome Informatics database (http://www.informatics.jax.org, retrieved May 2022).

**Hierarchical clustering** Hierarchical clustering and heatmap generation was performed using the ComplexHeatmap (Gu et al, 2016) package using Euclidean distance or one minus Pearson correlation distance, and Ward's linkage. Clustering was performed in a supervised manner using statistically differentially expressed genes or proteins belonging to a particular curated pathway, for example, *HALLMARK_MYC_TARGETS_V1*.

### Proteomics

Corrected reporter intensities were extracted from the proteinGroups table and further normalised in the same manner as the method underlying the RNA-seq DE analysis (Love et al, 2014). Briefly, for each gene *i*, its expression in a pseudo-reference sample was constructed as the geometric mean intensity across samples:

$$R_i = exp\left(\frac{1}{n}\sum_j log\left(I_{ij}\right)\right) = \left(\prod_j I_{ij}\right)^{1/n,}$$

and size factors were computed for each sample *j* as its median intensity:

$$S_j = \underset{i}{median}\frac{I_{ij}}{R_i}.$$

The normalised intensity for each gene *i* in sample *j* was then calculated as

$$N_{ij} = \frac{I_{ij}}{S_j}.$$

Protein differential expression (DE) analysis between GCN2-inhibited cultures and the DMSO control at 48 h after treatment was performed on protein groups detected in all 4 DMSO and all 3 GCN2iB samples. Welch's two-sided *t* tests were performed on the log$_2$-transformed normalised intensities; *P*-values were adjusted using the Benjamini–Hochberg multiple testing correction.

### Overrepresentation analysis

For the purpose of overrepresentation analyses, genes were classified as differentially expressed if adjusted *P* < 0.05; unless otherwise noted, no effect size (log fold change) threshold was applied.

Gene Ontology (GO) overrepresentation analyses were performed using the *enrichGO* function from the clusterProfiler package (v.3.18.1 [Yu et al, 2012]). Genes were indexed by their Ensembl identifier, and annotations were taken from the org.Hs.eg.db database in Bioconductor (v.3.12.0). Genes detected in each data set were used as background. Analyses were restricted to biological_process GO terms with between 10 and 500 constituents; adjusted *P*-values were corrected with the Benjamini–Hochberg procedure, and gene sets were reported with *P*- and *q*-value cut-offs of 0.05.

Similarly, KEGG overrepresentation analyses were performed using the *enrichKEGG* function from the clusterProfiler package (v.3.18.1 [Yu et al, 2012]). Genes were indexed by their NCBI gene ID; the conversion was performed using the biomaRt package

(v.2.46.3 [Durinck et al, 2009]) with the default "Ensembl" mart. Genes for which the conversion succeeded and that were detected in each data set were used as background. Local annotations were provided using the Bioconductor KEGG.db package (v.3.2.4). Analyses were restricted to KEGG pathways with between 10 and 500 constituents; adjusted *P*-values were corrected with the Benjamini–Hochberg procedure, and pathways were reported with *P*- and *q*-value cut-offs of 0.05.

### GSEA

GSEA was performed using the fgsea package (v.1.16.0 [Korotkevich et al, 2021 *Preprint*]). Annotations to MSigDB Hallmark categories (Liberzon et al, 2015) were provided by the msigdbr package (v.7.5.1). Genes were ordered by their log2 fold change values after shrinkage was applied (see section RNA sequencing). The *P*-value boundary was set to 0 during the GSEA calculations, but adjusted *P*-values were truncated in downstream plots.

### Identification of mRNA and protein groups

We used test statistics for GCN2iB versus DMSO at 48 h for both mRNA-seq and proteomics. mRNA and proteins were compared and stratified into 5 groups.

Group 1

 mRNA NS and protein NS

Group 2

 mRNA SS up-regulated and protein SS up-regulated
 mRNA SS up-regulated and protein NS, but up-regulated
 mRNA NS, but up-regulated and protein SS up-regulated

Group 3

 mRNA SS up-regulated and protein SS down-regulated
 mRNA SS up-regulated and protein NS, but down-regulated
 mRNA NS, but up-regulated and protein SS down-regulated

Group 4

 mRNA SS down-regulated and protein SS down-regulated
 mRNA SS down-regulated and protein NS, but down-regulated
 mRNA NS, but down-regulated and protein SS down-regulated

Group 5

 mRNA SS down-regulated and protein SS up-regulated
 mRNA NS, but down-regulated and protein SS up-regulated
 mRNA SS down-regulated and protein NS, but up-regulated

 Note 1: log$_2$FC, log$_2$ fold change.
 Note 2: NS, not statistically significant.
 Note 3: SS, statistically significant.
 Note 4: up-regulated, log$_2$FC > 0.
 Note 5: down-regulated, log$_2$FC < 0.
 Note 6: SS is defined as adjusted *P* < 0.05.

For each group, 1 to 5, mRNA and proteins were further categorised based on protein designations from Uhlén et al (2015), namely: SP, secreted; TM, transmembrane; IC, intracellular; SPTM, SP and TM. The presence of each mRNA/protein from each category

(1–5) and each subcategory (SP, TM, IC, SPTM) was checked in different Gene Ontology terms:

GO:0042254 Ribosome biogenesis
GO:0022613 Ribonucleoprotein complex biogenesis
GO:0000502 Proteasome complex
GO:0005839 Proteasome core complex
GO:0030198 Extracellular matrix organization
GO:0043062 Extracellular structure organization

### Reactome enrichment analysis

Pathway enrichment analysis was performed on genes passing adjusted $P < 0.05$ via the ReactomeF1 plugin in Cytoscape. Selected statistically significantly enriched pathways were then plotted in R as networks via the enrichplot and ReactomePA packages.

### Identification of druggable targets

Significantly (adjusted $P < 0.05$) deregulated proteins were cross-checked against potentially druggable proteins described in the Human Protein Atlas (https://www.proteinatlas.org/humanproteome/tissue/druggable).

### Statistical analysis

Statistical analysis and graphical representation were conducted with GraphPad Prism v.10 software. The statistical test employed for each experiment is stated in the corresponding figure legend. For all experiments, $P < 0.05$ was considered statistically significant.

## Data Availability

RNA sequencing, proteomic data, and the code used are available on GitHub (https://github.com/mromantr/An-ISR-independent-role-of-GCN2-prevents-excessive-ribosome-biogenesis-and-mRNA-translation).

## Supplementary Information

## Acknowledgements

We wish to acknowledge Sandra Loaiza and the John Goldman Centre for Cellular Therapy at Imperial College Healthcare National Health Service (NHS) Trust for providing human CD34$^+$ haematopoietic stem cells and mesenchymal stromal cells in collaboration with Imperial College Healthcare Tissue Bank (ICHTB). We thank Nicholas Crump and Jason Taslim for research support. We are thankful to the Protein Analysis Facility, Center for Integrative Genomics, Faculty of Biology and Medicine, University of Lausanne, for proteomic analysis, and to the University of Lausanne Genomic Facility for RNA-seq analysis (Lausanne, Switzerland). This study was funded by a Cancer Research UK Advanced Clinician Scientist Fellowship to HW Auner (C41494/A29035), a Cancer Research UK Small Molecule Drug Discovery Award to HW Auner (C41494/A27988) that is supported by the Stand Up To Cancer (SU2C) campaign for Cancer Research UK, Fondation ISREC, and an intramural research grant from the Department of Oncology at the Lausanne University Hospital (CHUV). GRM was funded by a Baxter Fellowship from the University of Dundee.

## Author Contributions

M Román-Trufero: conceptualization, investigation, formal analysis, visualization, and writing—original draft, review, and editing.
IT Kleijn: formal analysis and visualization.
K Blighe: formal analysis and visualization.
J Zhou: investigation.
P Saavedra-García: investigation.
A Gaffar: investigation.
M Christoforou: investigation.
A Bellotti: investigation.
J Abrahams: investigation.
A Atrih: investigation.
D Lamont: investigation.
M Gierlinski: formal analysis.
P Jayaprakash: investigation and formal analysis.
AM Michel: methodology.
EO Aboagye: conceptualization and methodology.
M Yuneva: conceptualization.
GR Masson: conceptualization, investigation, and formal analysis.
V Shahrezaei: conceptualization and formal analysis.
HW Auner: conceptualization, formal analysis, visualization, supervision, funding acquisition, and writing—original draft, review, and editing.

### Conflict of Interest Statement

HW Auner and M Román-Trufero have received research funding from Apollo Therapeutics. The other authors declare no competing interests.

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
