## [Reviewer comments · Life Science Alliance]

Life Science Alliance

An ISR-independent role of GCN2 prevents excessive ribosome biogenesis and mRNA translation

Monica Roman-Trufero, Istvan Kleijn, Kevin Blighe, Jinglin Zhou, Paula Saavedra-Garcia, Abigail Gaffar, Marilena Christoforou, Axel Bellotti, Joel Abrahams, Abdelmadjid Atrih, Douglas Lamont, Marek Gierlinski, Pooja Jayaprakash, Audrey Michel, Eric Aboagye, Mariia Yuneva, Glenn Masson, Vahid Shahrezaei, and Holger Auner

DOI: <https://doi.org/10.26508/lsa.202403014>

Corresponding author(s): Holger Auner, University Hospital of Lausanne

Review Timeline:	Submission Date:	2024-08-23
	Editorial Decision:	2024-10-11
	Revision Received:	2025-01-17
	Editorial Decision:	2025-02-07
	Revision Received:	2025-02-13
	Accepted:	2025-02-14

Transaction Report:

October 11, 2024

Re: Life Science Alliance manuscript #LSA-2024-03014-T

Prof. Holger W Auner
Lausanne University Hospital (CHUV)
Division of Haematology and Central Haematology Laboratory
Rue du Bugnon 46
Lausanne 1011
Switzerland

Dear Dr. Auner,

Thank you for submitting your manuscript entitled "An Integrated Stress Response-independent role of GCN2 prevents excessive ribosome biogenesis and mRNA translation" to Life Science Alliance. The manuscript was assessed by expert reviewers, whose comments are appended to this letter. We invite you to submit a revised manuscript addressing the Reviewer comments.

Thank you for this interesting contribution to Life Science Alliance. We are looking forward to receiving your revised manuscript.

Sincerely,

B. MANUSCRIPT ORGANIZATION AND FORMATTING:

Reviewer #1 (Comments to the Authors (Required)):

Following up on the initial idea that GCN2 regulates important cellular functions in an ISR-independent manner, the authors use various omics approaches to show that GCN2 prevents excessive ribosome biogenesis and protein synthesis under conditions of optimal nutrient availability. This function is distinct to the canonical ISR and is regulated at the translational level. It is unclear if the authors are suggesting that this ISR-independent role for GCN2 is a general phenomenon or is specific to Cancer cells that rely on GCN2. Whilst this manuscript describes interesting findings related to tumour growth and cancer cell survival; one criticism is that the findings are also largely descriptive and do not offer significant mechanistic insights.

A major weakness is that the majority of omics analyses have been performed in a GCN2-dependent cell line without comparison to GCN2 independent cell lines. The authors show that a subset of cancer cells depend on GCN2 in the absence of nutrient depletion, or other stressors known to activate GCN2. To understand the mechanistic basis, they carried out transcriptome and proteome analyses in the melanoma cell line A375 as a model for GCN2-dependent cells. To properly interpret this data, it would have been important to compare how GCN2 inhibition affects the transcriptome and proteome in primary cells that are not defined as GCN2-dependent. ISR-independent effects should still be detected and it would allow differentiation of any gene expression changes that are specific to the GCN2-dependent cell line.

The authors have used phosphorylation of S6K to measure the activity of mTORC1 (Fig. S2B). The interpretation is over-stated (eg referring to a "minute increase") given the interpretation is not matched to statistically significant changes in phosphorylation. It is also unclear why the authors use complete glutamine depletion (%Q) here compared with 12.% depletion in Fig. 2A and B.

The authors clustered RNA-seq data and concluded that GCN2-independent cell lines cluster differently from GCN2-dependent lines (Fig. S3). The actual data is more nuanced than the black and white description provided in the text. In fact, the two independent cell lines clearly cluster with one of the dependent cell lines. It is never really explained what the authors mean by independent, dependent and highly dependent cell lines. This needs much more explanation including data describing the quantitative differences underlying this categorisation.

Fig. S4 - The interpretation of puromycin labelling is over-stated based on the data presented. The authors state that they observed evidence for increased protein synthesis in GCN2-dependent cells but not in GCN2 independent cells. But there is no statistical difference in one of the two dependent cell lines (MiaPaCa2) tested making it more akin to the independent cell lines and contradicting this interpretation.

Other comments

Fig. 1A - needs more explanation. We are told that the 16 different solid tumour cell lines have varying degrees of GCN2 dependency without any further details. Does cell death correlate with the degree of Gcn2 dependency? More details are required describing the degree of GCN2-dependency in these cell lines.

Fig. 2. The authors should include GCN2 inhibition or knock-down in healthy primary cells to test whether there is a similar crease in global protein synthesis. This is an important control to show that it is not something specific to the A375 melanoma cell line.

Fig. 1B and Fig. S1A is missing a control to show that GCN2 inhibition does not trigger cell death in cell lines that are not defined as GCN2-dependent (eg healthy primary cells).

Fig. S1F, G - The data dose not match description provide in the text. A significant reduction in tumour growth was only observed following inhibitor treatment and not for GCN2 knock-down.

Reviewer #2 (Comments to the Authors (Required)):

In this manuscript, the authors reported transcriptional and translational changes in cells upon GCN2 inactivation, even under nutrient-rich conditions. Although GCN2 inactivation suppressed tumor growth, the myc program was upregulated. The authors

propose that dysregulated cellular metabolism likely contributed to the observed phenotype.

The study is based on comprehensive data sets ranging from transcriptome, translome, and proteome. The majority of the results are informative, and the discussion part is well-thought out. The increased ribosome occupancy in cells with GCN2 inhibition is intriguing, but alternative interpretation is possible (see below). My biggest concern is the so-called "an integrated stress response-independent role of GCN2". The ISR does not act as an "on-or-off" switch. Without amino acid deprivation, the basal levels of ISR could not be ignored. The basal ISR could restrict global protein synthesis that is unleashed upon GCN2 silencing. To truly demonstrate the ISR-independent role for GCN2, the ISR inhibitors could help. On the other hand, the observed increase in ribosome biogenesis should be insensitive to ISR inhibitors.

The increased puromycin labeling shown in Figure 2C and 2D is interesting but caution needs to be taken during interpretation. Besides increased protein synthesis, the increased puromycin signals could be due to translational slowdown, which is more consistent with the constrained energy source highlighted in the manuscript. This could also explain the ribosome profiling data as well as the suppressed tumor growth. To distinguish these possibilities, independent assays are needed, e.g., [35S] methionine labeling or AHA labeling assays.

Many thanks for inviting us to submit a revised manuscript. We appreciate the reviewers' thoughtful and well-considered suggestions, which we have addressed with a comprehensive set of additional experiments that support our original findings. We hope that the revised manuscript is now acceptable for publication in Life Science Alliance.

REVIEWER 1

Reviewer comment:

Following up on the initial idea that GCN2 regulates important cellular functions in an ISR-independent manner, the authors use various omics approaches to show that GCN2 prevents excessive ribosome biogenesis and protein synthesis under conditions of optimal nutrient availability. This function is distinct to the canonical ISR and is regulated at the translational level. It is unclear if the authors are suggesting that this ISR-independent role for GCN2 is a general phenomenon or is specific to cancer cells that rely on GCN2. Whilst this manuscript describes interesting findings related to tumour growth and cancer cell survival; one criticism is that the findings are also largely descriptive and do not offer significant mechanistic insights.

We thank the reviewer for the suggestion that it was not yet fully clear from the manuscript that the ISR-independent role of GCN2 is specific to cancer cells that rely on GCN2. Our data suggest that this role is indeed limited to GCN2-dependent cancer cells, although we cannot exclude that this function also operates in other cells we have not tested. The additional findings presented in the revised manuscript further clarify this point.

While the precise molecular mechanism through which GCN2 regulates translation in GCN2-dependent cells remains to be elucidated, our findings provide significant insight into its role in regulating translation in a subset of cancers. We demonstrate that GCN2 inhibition leads to a clear upregulation in the transcription and translation of ribosomal genes, a response that we do not observe in GCN2-independent cells and primary cells.

A major weakness is that the majority of omics analyses have been performed in a GCN2-dependent cell line without comparison to GCN2 independent cell lines. The authors show that a subset of cancer cells depend on GCN2 in the absence of nutrient depletion, or other stressors known to activate GCN2. To understand the mechanistic basis, they carried out transcriptome and proteome analyses in the melanoma cell line A375 as a model for GCN2-dependent cells. To properly interpret this data, it would have been important to compare how GCN2 inhibition affects the transcriptome and proteome in primary cells that are not defined as GCN2-dependent. ISR-independent effects should still be detected, and it would

allow differentiation of any gene expression changes that are specific to the GCN2-dependent cell line.

To address this, we have now conducted RNA-seq and proteomic analyses of human umbilical vein endothelial cells (HUVECs), as well as proteomic analysis of two GCN2-independent cancer cell lines (HepG2 and IPC298) treated with GCN2iB for 48h (for which RNA-seq had already been done). Clustering based on groups of translation-related genes revealed upregulation of transcripts coding for ribosomal proteins, MYC targets, and translation initiation factors in GCN2-dependent but not in GCN2-independent cells or primary cells (Fig. S3B-E). Importantly, the main findings from these new experiments are that GCN2 inhibition has minimal effects on the proteome of primary cells and GCN2-independent cancer cells, with only a very small number of significantly deregulated proteins. This precludes any functionally meaningful analysis such as GSEA on those proteome data sets. The reviewer suggests that "ISR-independent effects should still be detected". However, the fact that we do not observe these effects in primary healthy cells and in GCN2-independent cancer cells supports the concept that the observed increase in ribosome biogenesis and translation is specific to GCN2-dependent cancer cells. The new results are included and shown in Supplementary Figures 3, 5 and 6, and described on pages 8 and 9.

The authors have used phosphorylation of S6K to measure the activity of mTORC1 (Fig. S2B). The interpretation is over-stated (eg referring to a "minute increase") given the interpretation is not matched to statistically significant changes in phosphorylation. It is also unclear why the authors use complete glutamine depletion here compared with 12.5% depletion in Fig. 2A and B.

In response to the reviewer's concerns, we have repeated the experiment by growing A375 cells in medium containing 12.5% glutamine to ensure the data are more directly comparable to those presented in Figures 2A and 2B. The updated results, now included as Supplementary Figure 2D, demonstrate that GCN2iB does not affect mTORC1 activity when cells are grown in complete medium. However, GCN2 inhibition prevents the expected repression of mTORC1 signalling that occurs when cells experience amino acid deprivation.

The authors clustered RNA-seq data and concluded that GCN2-independent cell lines cluster differently from GCN2-dependent lines (Fig. S3). The actual data is more nuanced than the black and white description provided in the text. In fact, the two independent cell lines clearly cluster with one of the dependent cell lines. It is never really explained what the authors mean by independent, dependent and highly dependent cell lines. This needs much

more explanation including data describing the quantitative differences underlying this categorisation.

To address this point and clarify the observations, we have undertaken two actions. First, we have incorporated RNA-seq data obtained from the GCN2 inhibition experiment performed in HUVECs to the clustering analysis. Second, we have rewritten the text to clarify that the clustering of the A549 cell lines closer to the independent cell is likely due to its lower sensitivity to GCN2iB. As detailed in the revised manuscript (Fig 1A and page 5), A549 cells are classified as “dependent”, however, their average survival of 89% after GCN2iB treatment places them just above the threshold we have defined for GCN2-independent cells. This low level of dependency is therefore likely to account for their “proximity” to the GCN2-independent cancer cells and primary cells in the clustering analysis.

Fig. S4 - The interpretation of puromycin labelling is over-stated based on the data presented. The authors state that they observed evidence for increased protein synthesis in GCN2-dependent cells but not in GCN2 independent cells. But there is no statistical difference in one of the two dependent cell lines (MiaPaCa2) tested making it more akin to the independent cell lines and contradicting this interpretation.

We have revised the text to clarify that highly GCN2-dependent cell lines (A375 and A2058) exhibit a statistically significant increase in protein synthesis, while dependent cells (MiaPaCa2) also show a clear increase in puromycin incorporation, although it does not reach statistical significance (page 9). This observation places these cells between independent and highly dependent cells and thus supports our conclusions. Lastly, and importantly, GCN2iB does not affect global protein synthesis in GCN2-independent cancer cell lines (IPC298 and HepG2) or in primary healthy cells (also refer to the comment below on Fig.2).

Other comments

Fig. 1A - needs more explanation. We are told that the 16 different solid tumour cell lines have varying degrees of GCN2 dependency without any further details. Does cell death correlate with the degree of Gcn2 dependency? More details are required describing the degree of GCN2-dependency in these cell lines.

We have updated Fig. 1A and revised the text (page 5) to provide a clearer explanation of the results. Cancer cell lines have now been categorized as highly GCN2-dependent, GCN2-dependent, or GCN2-independent, with an explanation of the degree of sensitivity to GCN2iB for each category.

Fig. 2. The authors should include GCN2 inhibition or knock-down in healthy primary cells to test whether there is a similar crease in global protein synthesis. This is an important control to show that it is not something specific to the A375 melanoma cell line.

To address this important suggestion, we performed GCN2 inhibition in HUVECs and human dermal fibroblasts (HDFs) and measured puromycin incorporation. In both cases, we did not observe an increase in protein synthesis following 48h of GCN2iB treatment and these results have now been included in Fig. S5A and page 9. Together with our findings on protein synthesis in cancer cells lines, the data support the notion that changes in global protein synthesis occur in cancer cells that depend on GCN2.

Fig. 1B and Fig. S1A is missing a control to show that GCN2 inhibition does not trigger cell death in cell lines that are not defined as GCN2-dependent (eg healthy primary cells).

We acknowledge the reviewer's observation that generating a GCN2 KD would have been a valuable control. Unfortunately, within the time frame of this revision, it was not feasible to establish infection protocols for primary cells. Nevertheless, we have included additional GCN2 inhibition via GCN2iB in two more types of primary cells (HUVECs and HDFs) as discussed above. We believe these results provide robust evidence confirming that primary healthy cells are not dependent on GCN2 under conditions of full nutrient availability.

Fig. S1F, G - The data dose not match description provide in the text. A significant reduction in tumour growth was only observed following inhibitor treatment and not for GCN2 knock-down.

In response to the reviewer's comment and have now amended the text (page 6) to clarify that only GCN2iB leads to a statistically significant reduction in tumour size. However, the numerically highly similar reduction in tumour growth after GCN2 knock-down in vivo, even if it did not reach statistical significance, supports the notion that GCN2 is required for unimpeded tumour growth. We have amended the text to reflect this more clearly.

REVIEWER 2

In this manuscript, the authors reported transcriptional and translational changes in cells upon GCN2 inactivation, even under nutrient-rich conditions. Although GCN2 inactivation suppressed tumour growth, the myc program was upregulated. The authors propose that dysregulated cellular metabolism likely contributed to the observed phenotype.

The study is based on comprehensive data sets ranging from transcriptome, translome, and proteome. The majority of the results are informative, and the discussion part is well-thought out. The increased ribosome occupancy in cells with GCN2 inhibition is intriguing, but alternative interpretation is possible (see below). My biggest concern is the so-called "an integrated stress response-independent role of GCN2". The ISR does not act as an "on-or-off" switch. Without amino acid deprivation, the basal levels of ISR could not be ignored. The basal ISR could restrict global protein synthesis that is unleashed upon GCN2 silencing. To truly demonstrate the ISR-independent role for GCN2, the ISR inhibitors could help. On the other hand, the observed increase in ribosome biogenesis should be insensitive to ISR inhibitors.

We thank reviewer 2 for the thoughtful comments and carried out ISR inhibitor experiments based on culturing A375 cells in full medium in the presence of the ISR inhibitor ISRIB. Initially, we performed a dose-response experiment and observed no effect of ISRIB on the viability of A375 cells in complete medium, as shown in Fig. S2B. Subsequently, we cultured the cells with 1 μ M of ISRIB and conducted puromycinylation assays, which revealed no change in global protein synthesis (shown in Fig. 2E). Additionally, qRT-PCR analysis of key ISR genes, performed on RNA extracted from A375 cells grown in complete and glutamine-depleted medium, showed that ISRIB does not affect the expression of the analysed genes in complete medium, while it suppresses the predicted increase in expression induced by glutamine depletion. We also observed no effect on the expression of selected ribosomal protein genes. Thus, ISRIB suppresses the ISR that is triggered by glutamine depletion but has no relevant effect on cell viability, global protein synthesis, or the ISR when cells are cultured under nutrient-rich conditions, in line with the notion that GCN2 dependency is not linked to ISR dependency under such conditions. New expression data are not shown in Fig 2F and Supp Fig 2C and described on pages 7 and 8. In relation to the comment concerning "basal levels of ISR", we considered and discussed this as part of the original submission. An effect of GCN2 inhibition or depletion on basal ISR levels that leads to increased protein synthesis would imply changes in eIF2 α phosphorylation and ISR gene expression. However, we show that GCN2 inhibition or depletion does not have a discernible effect on the ISR (Fig 2A, 2B, S2A), thus ruling out this possibility.

The increased puromycin labelling shown in Figure 2C and 2D is interesting but caution needs to be taken during interpretation. Besides increased protein synthesis, the increased puromycin signals could be due to translational slowdown, which is more consistent with the constrained energy source highlighted in the manuscript. This could also explain the ribosome profiling data as well as the suppressed tumour growth. To distinguish these

possibilities, independent assays are needed, e.g., [35S] methionine labelling or AHA labelling assays.

We appreciate the reviewer's comment, but the puromycinylation assay is widely considered to be a robust method for quantifying global protein synthesis. Puromycin incorporation reflects active synthesis of nascent polypeptides: as an analogue of aminoacyl-tRNA, puromycin is incorporated into elongating peptides. Therefore, higher puromycin incorporation indicates a higher rate of active translation as more peptides are available for incorporation. We would like to highlight that we have used a cycloheximide control in all our puromycinylation experiments. Cycloheximide blocks translation and triggers ribosome stalling, and the control confirms that stalled ribosomes do not incorporate puromycin. We are therefore confident that puromycin incorporation is a reliable and robust method to assess global rates of active translation that has largely replaced [35S] methionine labelling which is no longer feasible in many research centres.

The results of the new experiments for this revision have been incorporated into the discussion section. Specifically, the inhibition of the ISR via ISRIB has been discussed on page 14, while the inhibition of GCN2 in primary cells is addressed on page 15.

Finally, we have revised the Material and Methods section to reflect the new experiments we have performed (page 22) as well as modified table S1 with the additional cells used and added table S2 with the list of primers employed in this study.

February 7, 2025

RE: Life Science Alliance Manuscript #LSA-2024-03014-TR

Prof. Holger W Auner
University Hospital of Lausanne
Division of Haematology and Central Haematology Laboratory
Rue du Bugnon 46
Lausanne 1011
Switzerland

Dear Dr. Auner,

Thank you for submitting your revised manuscript entitled "An ISR-independent role of GCN2 prevents excessive ribosome biogenesis and mRNA translation". We would be happy to publish your paper in Life Science Alliance pending final revisions necessary to meet our formatting guidelines.

- please be sure that the authorship listing and order is correct
- please upload your main manuscript text as an editable doc file
- please upload all figure files as individual ones, including the supplementary figure files; all figure legends should only appear in the main manuscript file
- please add the X and Bluesky handles of your host institute/organization as well as your own or/and one of the authors in our system
- please note that titles in the system and manuscript file must match
- please add an Author Contributions section to your main manuscript text
- please add your main, supplementary figure, and table legends to the main manuscript text after the references section;
- please add a callout for Figures S6A-D and S7A-C to your main manuscript text
- please update the Data Availability statement to include the accession information for datasets and code

FIGURE CHECK:

- please add sizes next to all blots

A. FINAL FILES:

B. MANUSCRIPT ORGANIZATION AND FORMATTING:

Sincerely,

Reviewer #1 (Comments to the Authors (Required)):

I am happy that all the requested changes to the manuscript have been made and believe that this manuscript is now suitable for publication in LSA

Reviewer #2 (Comments to the Authors (Required)):

In this revised manuscript, the authors conducted some additional experiments to address the reviewers' concern. Although the new result using ISRIB is appreciative, the authors argument about puromycin labeling is totally wrong. Although it is understandable that authors are hesitant to conduct [35]S metabolic labeling, AHA labeling is well-established and straightforward. Without independent validation of global protein synthesis, the manuscript falls short of rigor required for any journals.

February 14, 2025

RE: Life Science Alliance Manuscript #LSA-2024-03014-TRR

Prof. Holger W Auner
University Hospital of Lausanne
Division of Haematology and Central Haematology Laboratory
Rue du Bugnon 46
Lausanne 1011
Switzerland

Dear Dr. Auner,

Thank you for submitting your Research Article entitled "An ISR-independent role of GCN2 prevents excessive ribosome biogenesis and mRNA translation". It is a pleasure to let you know that your manuscript is now accepted for publication in Life Science Alliance. Congratulations on this interesting work.

DISTRIBUTION OF MATERIALS:

Again, congratulations on a very nice paper. I hope you found the review process to be constructive and are pleased with how the manuscript was handled editorially. We look forward to future exciting submissions from your lab.

Sincerely,
